# Geometric Constraints for Small Language Models to Understand and Expand Scientific Taxonomies

**Liri Fang**[§]**, Dongqi Fu**[⋆]**, Jiawei Han**[§]**, Jingrui He**[§]**, Vetle I. Torvik**[§]

University of Illinois Urbana-Champaign[§], Meta AI[⋆]

{lirif2, hanj, jingrui, vtorvik}@illinois.edu, dongqifu@meta.com

https://github.com/LiriFang/SS-Mono

## Abstract

Recent findings reveal that token embeddings of Large Language Models (LLMs) exhibit strong hyperbolicity. This insight motivates leveraging LLMs for scientific taxonomy tasks, where maintaining and expanding hierarchical knowledge structures is critical. Although potential, generally-trained LLMs face challenges in directly handling domain-specific taxonomies, including computational cost and hallucination. Meanwhile, Small Language Models (SLMs) provide a more economical alternative if empowered with proper knowledge transfer. In this work, we introduce SS-Mono (**Structure-Semantic Monotonization**), a novel pipeline that combines local taxonomy augmentation from LLMs, self-supervised fine-tuning of SLMs with geometric constraints, and LLM calibration. Our approach enables efficient and accurate taxonomy expansion across root, leaf, and intermediate nodes. Extensive experiments on both leaf and non-leaf expansion benchmarks demonstrate that a fine-tuned SLM (e.g., DistilBERT-base-110M) consistently outperforms frozen LLMs (e.g., GPT-4o, Gemma-2-9B) and domain-specific baselines. These findings highlight the promise of lightweight yet effective models for structured knowledge enrichment in scientific domains.

## 1 Introduction

Recently, researchers discovered that token embeddings of Large Language Models (LLMs) can exhibit a high degree of hyperbolicity, which implies a latent hyperbolic structure in the embedding space (Patil et al., 2025; Yang et al., 2025). Building on this insight, fine-tuning LLMs in hyperbolic space could yield strong performance gains in an efficient manner (Yang et al., 2025). Similarly, this phenomenon is also verified, to some extent, that the embedding matrices of LLMs show the semantic structures, e.g., directions of antonym pairs (Kozlowski et al., 2025). Above evidence suggests that LLMs have the potential to be a powerful tool for solving the *scientific taxonomy* related tasks, like knowledge understanding and enrichment.

Scientific taxonomy, as a specific kind of text-attributed graph, in addition to the textual concept attached to each node, has a more rigorous and hierarchical structure than normal undirected graphs, i.e., which can be represented within an explicit hierarchy such as trees or directed acyclic graphs for the hypernym and hyponym, e.g., *Glycoproteins → Proteins → Ribosomal Proteins → Peptide Elongation Factors*, as shown in Figure 1. In the real world, scientific taxonomy is now serving many applications, such as knowledge organization and question answering (Shen & Han, 2022).

According to the above discussion, the hyperbolic space discovery in LLM's embedding space indicates the direction that LLM can solve the scientific taxonomy tasks. However, scientific taxonomy, as a type of controlled vocabulary, is always domain-specific, and LLMs' pre-training is usually executed on a large-scale general corpus. This disagreement means that highly likely LLMs can not be directly used for scientific taxonomies like prompting or in-context learning, but often call for the post-training or self-supervised fine-tuning process. Based on the recent studies (He et al., 2024), LLMs are not always affordable, especially when involved with fine-tuning, and under 'suitable operations' small language models (SLMs) can be sufficiently powerful and economical

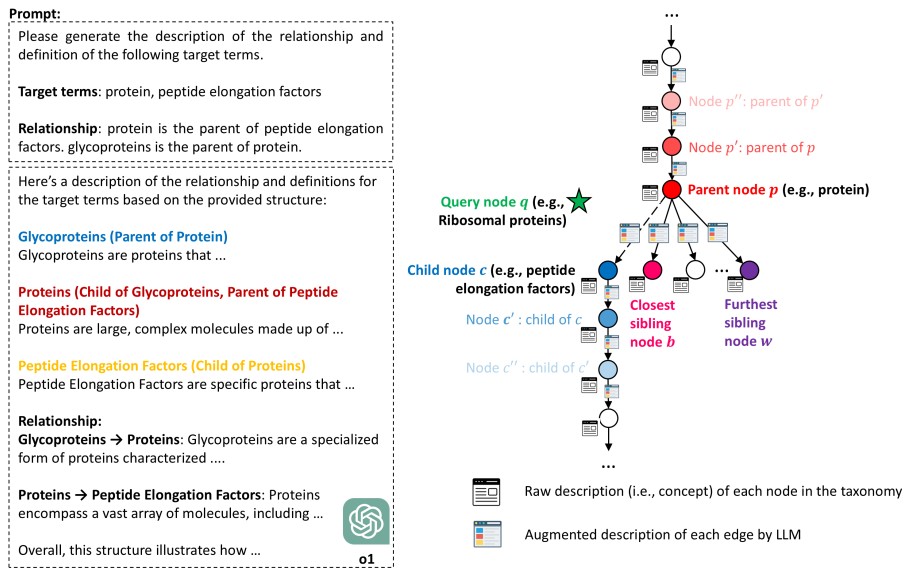

Figure 1: Scientific Taxonomy with LLM Augmentation (Edge-Level).

for many application scenarios and pave the way for the future of agentic AI (Belcak et al., 2025). Then, we need to ask, for the specific scientific taxonomy domain, are we able to first provide an LLM-to-SLM solution?

In the era of big data, new concepts continuously emerge, posing significant challenges for maintaining structured knowledge systems. *Taxonomy expansion* aims to insert the newly emerged concepts to the existing taxonomy appropriately instead of constructing a whole new taxonomy from scratch (Jiang et al., 2023; Zeng et al., 2024b; Xu et al., 2025). In this paper, we consider a more general and challenging taxonomy expansion problem, such that the query concept can be inserted everywhere in the existing taxonomy, including the root, leaf, and anywhere in between. As shown in Figure 3, the insertion in between is realized by the *Query-Position Matching* process: taking every existing edge as the candidate position (candidate answer) to a query, the query will rank all of them based on a scoring function, and select the highest rank to break its old edge and add two new edges. More details are also visualized in Figure 4 in Appendix L.1.

To begin with, we first verify that LLMs have great potential (and larger model performs better) but are not capable of directly understanding (or through simple prompting) the entire domain-specific taxonomy and making the correct expansion for the following reasoning and cases: (1) *Long Context Limit*: tested LLMs are incapable of taking entire existing text-attributed graph as input; (2) *Hallucination*: tested LLMs are prone to imagine non-existing edges in the existing taxonomy for query to insert; (3) *No Answer*: tested LLMs fail to generate available answer for the taxonomy expansion; (4) *Partial Answer*: tested LLMs only generate a part of correct answer. The real-world failed cases and statistics are shown in Section 4.3 and Figure 2.

Based on the above preliminary testing, we propose the design principle that: on the one hand, we need to 'borrow knowledge' from LLMs to SLMs; on the other hand, the 'borrowing' process should avoid computational cost as much as possible. Motivated by this, we propose the method named **SS-MONO** relying on (1) local taxonomy augmentation by an LLM, (2) fine-tuning of an SLM with geometric constraints, and (3) LLM calibration. The above pipeline strictly follows the existing hierarchical topology structure, considers the context of the raw textual attribute, adheres to augmentation by LLMs, and verifies the calibration of LLMs. We name this pipeline **Structure-Semantic Monotonization**. Empirically, the entire training process of SS-MONO is self-supervised. With leaf and non-leaf taxonomy expansion benchmark, a fine-tuned tiny LM like DistilBERT-base-110M leads the comprehensive outperformance over frozen general LLMs (like Gemma-2-9B (Mesnard et al., 2024) and GPT-4o mini (Hurst et al., 2024)) and domain-specific baselines.

## 2 PRELIMINARY

We define a taxonomy $\mathcal{T} = (V, E)$ as a directed acyclic graph (i.e., DAG), where each node $v \in V$ represents a unique concept, and a directed edge $(p, c) \in E$ represents a relation pointing from the parent node $p$ to the child node $c$. Furthermore, each concept (i.e., node $v \in V$) has a textual description, such that we can obtain the embedding features of each concept through language models. The corresponding feature matrix of the input taxonomy graph $\mathcal{T}$ (including the query node $q$) is denoted as $\mathbf{H} \in \mathbb{R}^{|V \cup \{q\}| \times h}$, where $h$ is the feature dimension, and we use $\mathbf{H}_v \in \mathbb{R}^h$ to denote the input feature vector of node $v$. The detailed process of obtaining $\mathbf{H}$ from fine-tuning language models can be found in Appendix L.2.

## 3 PROPOSED SS-MONO

In this section, we start to introduce the proposed framework SS-MONO, whose core technique is named structure-semantic monotonization. Here, we first introduce the overview and then use three subsections to illustrate the implementation details systematically.

### 3.1 OVERVIEW OF STRUCTURE-SEMANTIC MONOTONIZATION

Based on the existing taxonomy $\mathcal{T} = (V, E)$, the core of SS-MONO is to explore and integrate the structural information and contextual semantics of concepts to seek the best candidate position to insert the new concept. To achieve this matching, SS-MONO relies on the proposed structure-semantic monotonization via two encoder modules: **structure-dominated encoder** introduced in Section 3.2 and **context-dominated encoder** introduced in Section 3.3.

First, the structure-dominated encoder tries to verify whether the query node posits in the correct position bounded by the positions of its ground-truth hypernym (i.e., parent node) and ground-truth hyponym (i.e., child node). In other words, their relationship should be monotonic along the taxonomy structure. To verify this, the structure-dominated encoder adapts the hyperbolic representation learning to project their contextualized embedding into a hyperbolic space so that their hyperbolic embeddings obey the monotonic relationship along the taxonomy, i.e., **the transitivity in the hyperbolic space**. With this kind of hyperbolic embedding, we can try to compute the corresponding ranking score to rank the candidate positions for matching the query concept.

However, the contextual semantics in a certain taxonomy are limited compared with the large language models. Therefore, we propose the second module, context-dominated encoder. Intuitively, this encoder tests whether the semantic meaning around a candidate position shares the similarity with the query node. To obtain the semantic meaning of a candidate position, a frozen LLM is first prompted to give the textual explanation. Then the context-dominated encoder samples ancestors, descendants, and siblings along the hierarchy from that candidate position, encodes the text (augmented and raw node textual attribute) into representation vectors, and computes the matching score between the candidate position and the query.

To make these two encoder modules well-trained, we finally introduce self-supervised optimization, i.e., using the existing taxonomy to guide the learning process without human labeling costs.

### 3.2 STRUCTURE-DOMINATED ENCODER

Since taxonomy organizes concepts in the explicit hierarchy, this hierarchical structure restricts the concepts to follow a particular order from parent to child. Accordingly, the appropriate candidate position $(p, c)$ for a query $q$ to insert should satisfy the transitivity of hierarchical relations between position $(p, c)$ and query $q$, i.e., $c \preceq q \preceq p$.

To this end, SS-MONO preserves the hierarchical relation among concepts (e.g., query and candidate positions) together with their contextualized embedding $\mathbf{H}$ by adapting (1) hyperbolic encoding (Tifrea et al., 2019) to project $\mathbf{H}$ into a hierarchy-preserved metric space and (2) nested entailment cones (Ganea et al., 2018a) to regulate the projection to obey the hierarchical transitivity in the hyperbolic space. In the following two subsections, we first introduce the hyperbolic embedding method and then explain why the transitivity is preserved.

**Hyperbolic Encoding** Given the contextualized embedding $\mathbf{H} \in \mathbb{R}^{|\tilde{V} \cup q| \times h}$, in order to preserve their hierarchical relationships, the first step is to project $\mathbf{H}$ into a hyperbolic space, because the hyperbolic space fits the tree-like structure more for providing more space for lower level entries than Euclidean space (Tifrea et al., 2019; Chami et al., 2020).

Mathematically, we use Poincaré ball, one model of hyperbolic space. To be specific, the space is defined as $\mathbb{D}^d = \{x \in \mathbb{R}^d : \|x\| < 1\}$, with the Riemannian metric $g_x^{\mathbb{D}} = \lambda_x^2 g^{\mathbb{E}}$, where $\lambda_x := \frac{2}{1-\|x\|^2}$, $g^{\mathbb{E}} = \mathbf{I}_n$ and $\| \cdot \|$ is the Euclidean norm. Then, based on (Ganea et al., 2018b), two necessary transformation operations between this Poincaré ball and Euclidean space, map from transformer embeddings (Euclidean space) to hyperbolic space or vice versa.

Therefore, in SS-MONO, we can project the contextualized embedding $\mathbf{H}$ to the Poincaré ball space by a linear map and exponential map at the origin point 0. For each node $p$, let $z_p = \mathbf{W}^\top \mathbf{H}_p \in T_0 \mathbb{D}^d$, where $\mathbf{H}_p \in \mathbb{R}^h$ and $\mathbf{W} \in \mathbb{R}^{h \times d}$. Then, $\mathbf{H}'_p = \exp_0(z_p)$, where $\exp_0(\cdot)$ is the exponential map function with detailed computations illustrated in Appendix C.

**Nested Entailment Cones** To regulate the transitivity of hyperbolic embeddings, nested entailment cones (Ganea et al., 2018a) are adapted in SS-MONO.

*Claim* 3.1. Given two hierarchical relationships $(p, q)$ and $(q, c)$, the angle between $p$ and $q$ should be smaller than the half aperture of the parent cone $\mathfrak{S}_p^{\phi(p)}$, and the angle between $q$ and $c$ should be smaller than the half aperture of the query cone $\mathfrak{S}_q^{\phi(q)}$.

By introducing a cone $\mathfrak{S}_u^{\phi(u)}$ of a point $u$ with the width function $\phi(u)$ that satisfies the transitivity of partial order in an embedding space (as described in Appendix D), the ultimate goal is to ensure that SS-MONO regularizes hierarchical relation in the taxonomy obeying the angle $\Xi(u, v) \leq \phi(u)$ for pair $v \preceq u$ (i.e., $v$ is the child of $u$). To avoid the singularity at the origin and ensure the aperture $\phi(u)$ is well-defined, we restrict cone apices to $u \in \mathbb{D}^d \setminus B(0, \varepsilon)$ (i.e., $\|u\| \geq \varepsilon$).

Therefore, we design the energy score $E(u, v)$ based on cone modeling. Accordingly, the objective of cone loss is defined as a max-margin loss to enforce $E(u, v) = 0$ for positive examples (i.e., ground-truth matched query and position) and $E(u, v) > \gamma$ for negative ones.

$$E(u, v) := \max(0, \Xi(u, v) - \phi(u)) \tag{1}$$

The corresponding loss function is defined as follows.

$$\mathcal{L}_{\text{cone}}(u, v) = E(u, v) + \mathbb{E}_{(u^{(-)}, v^{(-)}) \sim \mathcal{N}(u,v)} \left[ \max \left( 0, \ \gamma - E(u^{(-)}, v^{(-)}) \right) \right], \tag{2}$$

where $\mathcal{N}(u, v)$ is negative-sampling distribution that returns a negative pair $(u^{(-)}, v^{(-)})$ different from $(u,v)$. Here, negative pairs refer to $u^{(-)}$ is not the ancestor of $v^{(-)}$. $\gamma > 0$ is the margin.

The structure loss $\mathcal{L}_{\text{structure}}$ is the summation of $\mathcal{L}_{\text{cone}}$ on $p$ and $q$ and on $q$ and $c$ for a given candidate position $(p, c)$ and a query node $q$.

$$\mathcal{L}_{\text{structure}} = \mathcal{L}_{\text{cone}}(\mathbf{H}'_p, \mathbf{H}'_q) + \mathcal{L}_{\text{cone}}(\mathbf{H}'_q, \mathbf{H}'_c) \tag{3}$$

where $\mathbf{H}'_p, \mathbf{H}'_q, \mathbf{H}'_c$ is the hyperbolic embedding of parent node $p$, query node $q$ and child node $c$, and the label generation is discussed in Section 3.4.

### 3.3 CONTEXT-DOMINATED ENCODER

Compared with LLMs, the semantics information in a certain taxonomy is not that rich. It is common to see only node has textual attributes but not edges (Bordea et al., 2016; Lipscomb, 2000; Jurgens & Pilehvar, 2016), and the construction of the existing taxonomy is often hand-crafted with no explicit knowledge to follow.

To provide enough context information, we first introduce a frozen LLM and prompt it with our designed template (details are in Appendix M), such that it can output the explanation of a candidate position $(p, c)$ about why a directed edge connected the hypernym and hyponym in the existing taxonomy $\mathcal{T}$, as the example of ("*Protein*", "*Peptide Elongation Factors*") shown in the left of Figure 1.

For further collecting the contextualized semantics of $(p, c)$ from the given taxonomy $\mathcal{T}$, three kinds of relationships need to be considered for query $q$, i.e., its ancestors, descendants, and siblings. For example, the candidate position ("*Protein*", "*Peptide Elongation Factors*") is the appropriate position to insert query "*Ribosomal proteins*". Then after inserting, "*Protein*" becomes the parent of "*Ribosomal proteins*", "*Peptide Elongation Factors*" becomes the child of "*Ribosomal proteins*", other children of "*Protein*" become siblings of "*Ribosomal proteins*".

Next, we introduce the different aspects of context embedding manners respectively.

**LLM Guidance Encoding**  First, we have the augmented description of LLM towards a candidate position $(p, c)$. In order to force SS-MONO to fit the LLM's knowledge in an efficient way, this LLM is frozen, i.e., no fine-tuning is involved. Then, the representation vector of the augmented description, $\mathbf{R}_{\text{LLM}}$, is obtained.

$$\mathbf{R}_{\text{LLM}} = \text{SAM}\left[\mathbf{e}, \mathbf{H}_{\text{LLM}}\right] \tag{4}$$

where $\mathbf{H}_{\text{LLM}} = \text{PLM}(\text{LLM}(p, c))$ is a embedding vector. LLM stands for a frozen Large Language Model, e.g., Gemma (Mesnard et al., 2024) or Llama (Touvron et al., 2023), and $\text{LLM}(p, c)$ is the augmented description of the candidate position $(p, c)$ as shown in Figure 1. PLM stands for a frozen [1] relative small language model to get the embedding vector of text, which is a more affordable way to get the hidden representation vectors of text, like DistilBERT (Sanh et al., 2019).

Moreover, in Eq. 4, SAM stands for the self-attention mechanism (Vaswani et al., 2017), vector $\mathbf{e} \in \mathbb{R}^h$ is a randomized vector as the initial placeholder, its output after the self-attention mechanism serves as the relational vector $\mathbf{R}_{\text{LLM}}$.

**Ancestor Context Encoding**  This encoding method is proposed to project the contextualized embedding $\mathbf{H}_q \in \mathbb{R}^h$ of $q$ together with its ancestors into a semantic relational representation vector $\mathbf{R}_a$ as follows.

$$\mathbf{R}_a = \text{SAM}\left[\mathbf{e}, \mathbf{H}_{p''}, \mathbf{H}_{p'}, \mathbf{H}_p, \mathbf{H}_q\right] \tag{5}$$

where $\mathbf{R}_a$ means the semantic relational encoding with ancestors. Representation vector $\mathbf{H}_q$ is obtained through a fine-tuned SLM over the given textual attribute of node $q$. The details of the computation are shown in Appendix L.3, and the same manner applies to other text-attributed nodes in the existing taxonomy graph.

Eq. 5 is an instance containing 3-hop ancestors, given $p'$ is the parent of $p$, and $p''$ is the parent of $p'$. Note that, in DAG-based taxonomy, a node may have multiple parents. If so, multiple parents will be selected and concatenated.

**Descendant Context Encoding**  Similar to the ancestor context encoding, the descendant context encoding is defined as follows.

$$\mathbf{R}_d = \text{SAM}\left[\mathbf{e}, \mathbf{H}_q, \mathbf{H}_c, \mathbf{H}_{c'}, \mathbf{H}_{c''}\right] \tag{6}$$

where $\mathbf{R}_d$ is the semantic relational encoding with descendants. Eq. 6 is an instance containing 3-hop descendants, given $c''$ is the child of $c'$, and $c'$ is the child of $c$.

**Sibling Context Encoding**  For sibling context encoding, the token list formation is different from Eq. 5 and Eq 6. Because the taxonomy can be quite wide, i.e., a parent node can have various child nodes, which means the query $q$ can have multiple siblings when considering one candidate position. Beyond that, the meaning across the siblings can diverge and be dependent on the depth of the taxonomy. To this end, we borrow the philosophy from (Wang et al., 2022) to first sample the most similar sibling $s$ and the worst similar sibling $w$ in terms of the contextualized embedding $\mathbf{H}$ based on language models.

$$b = \text{argmax}_{v \in \text{Child}(p)} \text{CosSim}(\mathbf{H}_v, \mathbf{H}_q), \quad w = \text{argmin}_{v \in \text{Child}(p)} \text{CosSim}(\mathbf{H}_v, \mathbf{H}_q) \tag{7}$$

where $p$ is the parent node, $\text{Child}(p)$ is the set of all child nodes of $p$ besides $c$ in the existing taxonomy $\mathcal{T}$, and CosSim denotes the cosine similarity.

---

[1] Note that different from ancestor, descendant, and sibling context encodings, only in Eq 4, the SLM is frozen.

Then, the sibling semantics encoding can be expressed as follows.

$$\mathbf{R}_s = \text{SAM}\,[\mathbf{e}, \mathbf{H}_q, \mathbf{H}_b, \mathbf{H}_w] \tag{8}$$

Finally, with $\mathbf{R}_{\text{LLM}}, \mathbf{R}_a, \mathbf{R}_d, \mathbf{R}_s$, we can then sample training samples and design context-dominated loss function.

To be specific, when targeting candidate position samples, a positive position sample means the parent $p$, child $c$, and siblings $b$ and $w$ are **all** ground truth for the query $q$. Then, the straightforward idea is that a negative position sample means that **any** entry from $p$, $c$, $b$, and $w$ is not true towards $q$. To make positive samples obtain a higher context-based matching score $F(\cdot)$ and the negative samples take a lower score, we design the following loss function.

First, the context-based query-position matching score $F(\cdot)$ is expressed as follows.

$$F(\mathbf{H}_q, \mathbf{R}_{\text{LLM}}, \mathbf{R}_a, \mathbf{R}_d, \mathbf{R}_s) = \mathbf{W}_2(ReLU(\mathbf{W}_1\mathbf{R}_{\text{concat}} + \mathbf{b}_1) + \mathbf{b}_2) \tag{9}$$

where $\mathbf{W}_1$, $\mathbf{W}_2$, $\mathbf{b}_1$, and $\mathbf{b}_2$ are weight matrices obtained from trainable parameters, and $\mathbf{R}_{\text{concat}} = [\mathbf{H}_q; \mathbf{R}_{\text{LLM}}; \mathbf{R}_a; \mathbf{R}_d; \mathbf{R}_s]$ means concatenation of $\mathbf{H}_q$, $\mathbf{R}_{\text{LLM}}$, $\mathbf{R}_a$, $\mathbf{R}_d$, and $\mathbf{R}_s$. Then, the context-based loss function is designed as follows.

$$\mathcal{L}_{context} = -[y \log(F(\mathbf{H}_q, \mathbf{R}_{\text{LLM}}, \mathbf{R}_a, \mathbf{R}_d, \mathbf{R}_s)) + (1 - y) \log(1 - F(\mathbf{H}_q, \mathbf{R}_{\text{LLM}}, \mathbf{R}_a, \mathbf{R}_d, \mathbf{R}_s))] \tag{10}$$

where $y$ is the label for a candidate position, $y = 1$ means a positive position sample such that each entry from $(p, c, b, w)$ is ground truth towards $q$, and $y = 0$ means a negative position sample that anyone from $(p, c, b, w)$ is not the ground truth.

**Hard Training Samples**   Within a negative position sample, besides the scenario that every component is not true, the harder samples exist. For example, we can sample a candidate position $(p, \hat{c})$, where $p$ is the ground-truth parent for $q$, but $\hat{c}$ is not the ground-truth child for $q$. Similarly, we can also sample incorrect $\hat{p}, \hat{b}, \hat{w}$. Therefore, we further split Eq. 9 and Eq. 10 into a series of fine-grained computations for hard negative samples.

Just take $(p, c)$ and $(p, \hat{c})$ as an example, the fine-grained version of Eq. 9 targeting positive and negative descendants, $F_{desc}$, is expressed as follows.

$$F_{desc}(\mathbf{R}_d) = \mathbf{W}_4(ReLU(\mathbf{W}_3(\mathbf{R}_d) + \mathbf{b}_3) + \mathbf{b}_4) \tag{11}$$

where $\mathbf{W}_3$, $\mathbf{W}_4$, $\mathbf{b}_3$ and $\mathbf{b}_4$ are matrices of trainable parameters. Then, the corresponding context-based loss function Eq. 10 is specialized below.

$$\mathcal{L}_{context\_desc} = -[y \log(F_{desc}(\mathbf{R}_d)) + (1 - y) \log(1 - F_{desc}(\mathbf{R}_d))] \tag{12}$$

where $y = 1$ means the child position is the ground truth child to insert $q$, and $y = 0$ otherwise.

Follow the same way, we can design ancestor score $F_{anc}(\mathbf{R}_a)$ with ancestor loss $\mathcal{L}_{context\_anc}$ and sibling score $F_{sib}(\mathbf{R}_s)$ with sibling loss $\mathcal{L}_{context\_sib}$. Note that in $\mathcal{L}_{context\_sib}$, $y = 1$ iff two selected siblings are both ground truth.

## 3.4   Self-Supervised Optimization

To save human labeling efforts in the training SS-Mono, we introduce a self-supervised learning manner. The idea is straightforward. We first remove an existing concept from the existing taxonomy, then sample corresponding positive and negative samples to train SS-Mono, and test if SS-Mono could replace the removal correctly. Next, we introduce how the training samples are prepared and the entire loss function to train SS-Mono.

**Positive and Negative Sampling.** In the existing taxonomy, we select an existing transitive relation $(p, q, c)$, which means $p$ is the parent of $q$, and $q$ is the parent of $c$. Then, starting from $p$, we sample the best and least similar sibling for $q$ and get $b$ and $w$. Now, we have a positive sample $(p, c, b, w)$. For the negative sample, we randomly replace any component in $(p, c, b, w)$ with the rest nodes in the existing taxonomy. With the positive and negative samples, we trace the corresponding ancestors and descendants to compute the matching scores stated above. With those scores, we model the seeking of the best candidate position as a classification problem with the following loss function.

**Loss Function.** Below is the total loss function for training SS-MONO, which combines the individual loss based on structure and (fine-grained) context information, in a structure-semantic monotonization manner, as discussed above.

$$\mathcal{L}_{total} = \alpha\mathcal{L}_{structure} + \beta\mathcal{L}_{context} + \mu\mathcal{L}_{context\_desc} + \lambda\mathcal{L}_{context\_anc} + \xi\mathcal{L}_{context\_sib} \quad (13)$$

where $\alpha$, $\beta$, $\mu$, $\lambda$, and $\xi$ are hyperparameters to control the weights of individual loss functions.

## 4 EXPERIMENTS

### 4.1 DATASETS, BASELINES, AND METRICS

We prepared three public datasets, i.e., SemEval-Food, MeSH, and WordNet-Verb, as shown in Table 1. SemEval-Food is the taxonomy for the food domain, which is released by SemEval-2016 Task 13 (Bordea et al., 2016). MeSH contains the subgraph of the Medical Subject Headings (MeSH) in the biomedical domain, published by NLM annually (Lipscomb, 2000). WordNet-Verb is the verb taxonomy containing the description of each verb, which is published as SemEval 2016 Task 14 (Jurgens & Pilehvar, 2016).

We consider the leaf expansion and non-leaf expansion capabilities together. Therefore, we include the corresponding SOTA baselines: Bilinear Model (Sutskever et al., 2009), Neural Tensor Network (Socher et al., 2013), TaxoExpan (Shen et al., 2020), ARBORIST (Manzoor et al., 2020), TMN (Zhang et al., 2021), QEN (Wang et al., 2022), TaxBox (Xue et al., 2024). A more detailed introduction of baselines is placed in Appendix E.

Table 1: Dataset statistics. $|N|$, $|E|$, $D$, $|L|$, $L\%$, and $|Q|$ denote number of nodes, edges, depth, leaf nodes, leaf ratio, and query concepts.

| Dataset | $|N|$ | $|E|$ | $D$ | $|L|$ | $L\%$ | $|Q|$ |
|---|---|---|---|---|---|---|
| SemEval-Food | 1,486 | 1,533 | 8 | 1,184 | 79.7% | 148 |
| MeSH | 9,710 | 10,498 | 10 | 6,613 | 68.1% | 819 |
| WordNet-Verb | 13,936 | 13,407 | 12 | 10,581 | 75.9% | 1,000 |

Furthermore, we explore the ability of several LLMs (>1B) to retrieve and rank candidate edges as LLM baselines, including DeepSeek-R1-8B (DeepSeek-AI et al., 2025), Llama-3.1-8B (Touvron et al., 2023), Gemma-2-9B (Mesnard et al., 2024), and GPT-4o mini (Hurst et al., 2024). The implementation details of LLM baselines are provided in Appendix F. We prepared 15 metrics to comprehensively evaluate the performance of all baseline methods, covering recall, precision, mean, etc. The details of the illustration are in Appendix G. The generation and verification process of the augmented edge description by LLMs is given in Appendix M to demonstrate the trustworthiness of augmentation.

### 4.2 EFFECTIVENESS OF SS-MONO

Table 2 reports the comprehensive performance of all baselines on the SemEval-Food, WordNet-Verb, and MeSH datasets. SS-MONO (w/o AD) denotes the proposed model without LLM **a**ugmented **d**escription for every candidate position, and SS-MONO denotes the full proposed model. To be specific, LLM baselines like DeepSeek-R1-8B (DeepSeek-AI et al., 2025) or GPT-4o mini (Hurst et al., 2024) directly infer to rank the top 10 positions (Detailed implementation is in Appendix F, the analysis of the cardinality of candidate pool as the input of LLMs can be found in Appendix I.1). Consequently, if the ground truth edges do not appear among the top 10 candidates, we cannot compute rank-based metrics such as MR and MRR. The symbol – in Table 2 denotes cases where metric results are unavailable. TaxBox (Xue et al., 2024) does not report MR and R@10 for both leaf and non-leaf nodes, nor does it provide results for the MeSH dataset. Since TaxBox does not publicly release its implementation scripts, we mark its performance as –.

In general, as shown in Table 2, SS-MONO (w/o AD) achieves competitive performance compared with baselines, and SS-MONO achieves the best performance overall comparisons in each dataset. The corresponding visualization case study is placed in Appendix H. For cross-dataset comparisons of an individual algorithm, MRR (Mean Reciprocal Rank) and R@k (Recall@k) are more appropriate indicators, as they are scale-invariant and reflect relative ranking quality independent of taxonomy size. MR is absolute value grows naturally with larger and deeper taxonomies and cannot be directly compared across datasets with divergent scales. Also, in Table 2, it can be observed that including LLM Augmented Descriptions (AD) does not always enhance intermediate (non-leaf) expansion, a detailed analysis is placed in Appendix O. Moreover, we also prepared the performance of Fine-Tuned

Table 2: Performance comparison on three taxonomy expansion benchmarks. **Bold colors** indicate top-3 per column (best=red, second=blue, third=green). SS-Mono is our full model; SS-Mono (w/o AD) disables LLM augmentation.

**SemEval-Food**

| Type | Method | Total | | | | | | | | Leaf | | | Non-leaf | | |
|---|---|---|---|---|---|---|---|---|---|---|---|---|---|---|---|
| | | MR ↓ | MRR ↑ | R@1 ↑ | R@5 ↑ | R@10 ↑ | P@1 ↑ | P@5 ↑ | P@10 ↑ | MR ↓ | MRR ↑ | R@10 ↑ | MR ↓ | MRR ↑ | R@10 ↑ |
| LLM | DeepSeek-R1-8B | – | – | 0.016 | 0.016 | 0.016 | 0.033 | 0.007 | 0.003 | – | – | 0.028 | – | – | 0.005 |
| | Llama-3.1-8B | – | – | 0.003 | 0.006 | 0.006 | 0.007 | 0.003 | 0.001 | – | – | 0.007 | – | – | 0.006 |
| | Gemma-2-9B | – | – | 0.000 | 0.000 | 0.000 | 0.000 | 0.000 | 0.000 | – | – | 0.000 | – | – | 0.000 |
| | GPT-4o mini | – | – | 0.016 | 0.055 | 0.058 | 0.034 | 0.023 | 0.012 | – | – | 0.000 | – | – | 0.103 |
| Non-LLM | Bilinear | 700.07 | 0.140 | 0.024 | 0.096 | 0.110 | 0.050 | 0.039 | 0.022 | 269.89 | 0.305 | 0.244 | 2816.53 | 0.005 | 0.000 |
| | NTN | 685.41 | 0.192 | 0.037 | 0.102 | 0.148 | 0.074 | 0.041 | 0.030 | 241.65 | 0.422 | 0.328 | 2868.68 | 0.005 | 0.000 |
| | TaxoExpan | 688.70 | 0.207 | 0.041 | 0.101 | 0.166 | 0.083 | 0.041 | 0.034 | 255.64 | 0.455 | 0.368 | 2819.36 | 0.004 | 0.000 |
| | ARBORIST | 700.79 | 0.129 | 0.013 | 0.053 | 0.088 | 0.027 | 0.022 | 0.018 | 260.38 | 0.280 | 0.195 | 2867.65 | 0.005 | 0.000 |
| | TMN | 559.81 | 0.221 | 0.037 | 0.113 | 0.160 | 0.074 | 0.046 | 0.032 | 179.46 | 0.482 | 0.356 | 2431.13 | 0.007 | 0.000 |
| | QEN | 397.77 | 0.315 | 0.071 | 0.164 | 0.228 | 0.149 | 0.069 | 0.048 | 275.07 | 0.367 | 0.276 | 1230.86 | 0.099 | 0.033 |
| | TaxBox | 281.00 | 0.359 | 0.132 | 0.264 | 0.295 | 0.318 | 0.127 | 0.071 | – | 0.678 | – | – | 0.133 | – |
| Ours | SS-Mono (w/o AD) | 315.79 | 0.430 | 0.161 | 0.283 | 0.338 | 0.338 | 0.119 | 0.071 | 228.18 | 0.690 | 0.642 | 768.47 | 0.225 | 0.098 |
| | SS-Mono | 239.17 | 0.400 | 0.186 | 0.299 | 0.325 | 0.392 | 0.126 | 0.068 | 143.94 | 0.705 | 0.645 | 756.73 | 0.147 | 0.059 |

**WordNet-Verb**

| Type | Method | Total | | | | | | | | Leaf | | | Non-leaf | | |
|---|---|---|---|---|---|---|---|---|---|---|---|---|---|---|---|
| | | MR ↓ | MRR ↑ | R@1 ↑ | R@5 ↑ | R@10 ↑ | P@1 ↑ | P@5 ↑ | P@10 ↑ | MR ↓ | MRR ↑ | R@10 ↑ | MR ↓ | MRR ↑ | R@10 ↑ |
| LLM | DeepSeek-R1-8B | – | – | 0.000 | 0.000 | 0.000 | 0.000 | 0.000 | 0.000 | – | – | 0.000 | – | – | 0.000 |
| | Llama-3.1-8B | – | – | 0.000 | 0.000 | 0.000 | 0.000 | 0.000 | 0.000 | – | – | 0.000 | – | – | 0.000 |
| | Gemma-2-9B | – | – | 0.000 | 0.000 | 0.000 | 0.000 | 0.000 | 0.000 | – | – | 0.000 | – | – | 0.000 |
| | GPT-4o mini | – | – | 0.001 | 0.001 | 0.001 | 0.001 | 0.000 | 0.000 | – | – | 0.000 | – | – | 0.002 |
| Non-LLM | Bilinear | 1861.30 | 0.174 | 0.012 | 0.052 | 0.095 | 0.018 | 0.016 | 0.014 | 888.55 | 0.247 | 0.140 | 5851.59 | 0.089 | 0.044 |
| | NTN | 1568.62 | 0.251 | 0.050 | 0.124 | 0.171 | 0.075 | 0.037 | 0.026 | 819.93 | 0.413 | 0.309 | 4639.76 | 0.067 | 0.013 |
| | TaxoExpan | 2023.85 | 0.231 | 0.053 | 0.122 | 0.168 | 0.080 | 0.037 | 0.025 | 1127.28 | 0.392 | 0.308 | 5701.62 | 0.048 | 0.007 |
| | ARBORIST | 1499.40 | 0.238 | 0.033 | 0.096 | 0.149 | 0.049 | 0.028 | 0.023 | 838.69 | 0.315 | 0.204 | 4209.64 | 0.149 | 0.086 |
| | TMN | 1510.17 | 0.291 | 0.066 | 0.154 | 0.207 | 0.099 | 0.047 | 0.031 | 751.15 | 0.439 | 0.342 | 4623.67 | 0.121 | 0.052 |
| | QEN | 1802.40 | 0.340 | 0.081 | 0.186 | 0.249 | 0.124 | 0.057 | 0.038 | 1055.87 | 0.495 | 0.407 | 4909.49 | 0.166 | 0.093 |
| | TaxBox | 1286.00 | 0.330 | 0.105 | 0.212 | 0.262 | 0.179 | 0.072 | 0.045 | – | 0.481 | – | – | 0.185 | – |
| Ours | SS-Mono (w/o AD) | 2579.88 | 0.297 | 0.048 | 0.134 | 0.205 | 0.074 | 0.041 | 0.031 | 1746.02 | 0.373 | 0.296 | 6089.03 | 0.208 | 0.099 |
| | SS-Mono | 1626.52 | 0.334 | 0.106 | 0.208 | 0.260 | 0.163 | 0.064 | 0.040 | 922.54 | 0.521 | 0.457 | 4551.31 | 0.122 | 0.035 |

**MeSH**

| Type | Method | Total | | | | | | | | Leaf | | | Non-leaf | | |
|---|---|---|---|---|---|---|---|---|---|---|---|---|---|---|---|
| | | MR ↓ | MRR ↑ | R@1 ↑ | R@5 ↑ | R@10 ↑ | P@1 ↑ | P@5 ↑ | P@10 ↑ | MR ↓ | MRR ↑ | R@10 ↑ | MR ↓ | MRR ↑ | R@10 ↑ |
| LLM | DeepSeek-R1-8B | – | – | 0.003 | 0.005 | 0.008 | 0.006 | 0.002 | 0.002 | – | – | 0.011 | – | – | 0.005 |
| | Llama-3.1-8B | – | – | 0.001 | 0.001 | 0.001 | 0.001 | 0.000 | 0.000 | – | – | 0.002 | – | – | 0.002 |
| | Gemma-2-9B | – | – | 0.003 | 0.006 | 0.012 | 0.006 | 0.003 | 0.003 | – | – | 0.013 | – | – | 0.010 |
| | GPT-4o mini | – | – | 0.000 | 0.001 | 0.003 | 0.000 | 0.000 | 0.001 | – | – | 0.004 | – | – | 0.000 |
| Non-LLM | Bilinear | 985.23 | 0.273 | 0.038 | 0.115 | 0.173 | 0.086 | 0.052 | 0.039 | 483.02 | 0.395 | 0.284 | 2064.97 | 0.192 | 0.100 |
| | NTN | 702.32 | 0.329 | 0.064 | 0.167 | 0.227 | 0.143 | 0.075 | 0.051 | 408.17 | 0.542 | 0.454 | 1334.75 | 0.189 | 0.077 |
| | TaxoExpan | 6784.30 | 0.173 | 0.024 | 0.085 | 0.123 | 0.053 | 0.028 | 0.038 | 466.75 | 0.434 | 0.310 | 20367.05 | 0.001 | 0.000 |
| | ARBORIST | 800.81 | 0.173 | 0.024 | 0.085 | 0.123 | 0.053 | 0.028 | 0.038 | 466.75 | 0.434 | 0.310 | 1413.43 | 0.292 | 0.175 |
| | TMN | 494.31 | 0.410 | 0.061 | 0.197 | 0.291 | 0.137 | 0.088 | 0.065 | 401.70 | 0.555 | 0.459 | 693.42 | 0.315 | 0.180 |
| | QEN | 530.83 | 0.423 | 0.071 | 0.198 | 0.294 | 0.165 | 0.091 | 0.066 | 511.93 | 0.548 | 0.427 | 573.01 | 0.322 | 0.187 |
| | TaxBox | – | – | – | – | – | – | – | – | – | – | – | – | – | – |
| Ours | SS-Mono (w/o AD) | 584.68 | 0.408 | 0.048 | 0.175 | 0.267 | 0.112 | 0.082 | 0.063 | 602.11 | 0.479 | 0.363 | 546.99 | 0.365 | 0.209 |
| | SS-Mono | 436.82 | 0.427 | 0.074 | 0.197 | 0.288 | 0.173 | 0.093 | 0.068 | 390.72 | 0.570 | 0.476 | 540.55 | 0.334 | 0.166 |

LLM for the taxonomy expansion task in Appendix N. This experiment further confirms that (1) our geometric deep learning objective is easily compatible with off-the-shelf LLM checkpoints, i.e., only minimal modifications are needed to plug in an LLM encoder, and training remains computationally lightweight. (2) Fine-tuning LLMs is not always outperforming, the way we designed to "borrow" knowledge from LLMs to SLMs is effective and competitive.

## 4.3 CALIBRATION BY LLMS

Given a query $q$, SS-Mono will rank all the existing edges in the taxonomy and select the highest one to insert. Therefore, when SS-Mono outputs the ranking list, we insert this ranking list to a promoted LLM (a template example is given in Appendix J) and ask LLM to rerank it to the best of their knowledge.

For example, in the testing set of SemEval-Food, we have 148 queries to be inserted into the existing taxonomy, and the existing taxonomy has 7,313 candidate positions. In other words, for each one of 148, SS-Mono provides a ranking list of 7,313 entries, and Llama3.1-8B (Touvron et al., 2023) reranks them. Due to the long context limit of LLMs, we need to truncate the ranking list and ask Llama to only rerank the truncated list and leave the rest remaining. We use $k$ to denote the length of the truncated ranking list, e.g., $k = 10, 50, 100, 200$. Then, we evaluate the rerank (calibrated) ranking list and report the comparison in Table 3.

Table 3: Performance comparison of SS-MONO variants with LLM calibration on SemEval-Food. SS-MONO-$k$ denotes reranking the top-$k$ candidates using LLMs. **Bold** indicates the best score.

| Method | Total | | | | | | | Leaf | | | Non-leaf | | | Avg. Rank |
|---|---|---|---|---|---|---|---|---|---|---|---|---|---|---|
| | MR ↓ | MRR ↑ | R@1 ↑ | R@5 ↑ | R@10 ↑ | P@1 ↑ | P@10 ↑ | MR ↓ | MRR ↑ | R@10 ↑ | MR ↓ | MRR ↑ | R@10 ↑ | |
| SS-MONO (w/o AD) | 315.79 | 0.430 | 0.161 | 0.283 | 0.338 | 0.338 | 0.071 | 228.18 | 0.690 | 0.642 | 768.47 | 0.225 | 0.098 | 4.786 |
| SS-MONO | 239.17 | 0.400 | 0.186 | 0.299 | 0.325 | 0.392 | 0.068 | 143.94 | 0.705 | 0.645 | 756.74 | 0.147 | 0.059 | 4.643 |
| SS-MONO-10 | 240.07 | 0.398 | 0.138 | 0.235 | 0.322 | 0.291 | 0.067 | 139.70 | 0.721 | 0.657 | 758.65 | 0.144 | 0.057 | 5.071 |
| SS-MONO-50 | 238.13 | 0.439 | 0.203 | 0.334 | 0.373 | 0.426 | 0.078 | 138.18 | **0.736** | **0.679** | 754.52 | 0.205 | 0.132 | 2.143 |
| SS-MONO-100 | **237.46** | **0.462** | **0.206** | **0.350** | **0.389** | **0.432** | **0.082** | **138.06** | 0.727 | 0.664 | **751.02** | **0.253** | **0.172** | **1.357** |
| SS-MONO-200 | 238.06 | 0.417 | 0.190 | 0.318 | 0.341 | 0.399 | 0.072 | 137.99 | 0.728 | 0.664 | 755.05 | 0.171 | 0.086 | 2.929 |

██ Output without reranked list   ██ Output with hallucinated edges   ██ Output with shorter list

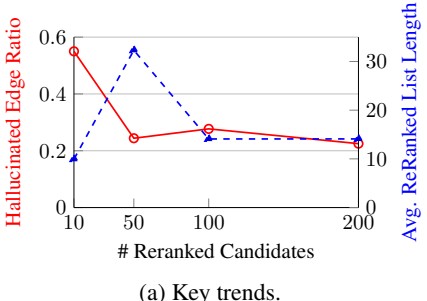
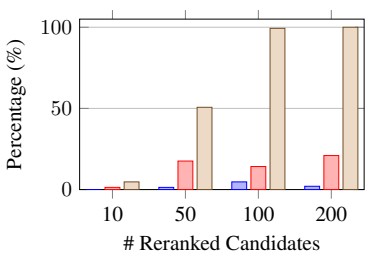

(a) Key trends.

(b) Failure-type percentages.

Figure 2: Statistics of failed cases in LLM calibration under different reranked candidate sizes. (a) Key trends: hallucinated edge ratio (red solid line) vs. average reranked list length (blue dash line). (b) Distribution across failure types, denoted by blue bar, red bar, and brown bar.

During the calibration, we also observe a considerable amount of failed cases of LLM's output, such as (1) demonstrated LLMs could not rerank the given ranking list but generate the rerank idea or python code; (2) demonstrated LLMs generate some not existing edges in the given ranking list, i.e., hallucination; (3) demonstrated LLMs are sometimes lazy to generate the full ranking list as the given. The statistics are shown in Figure 2, and concrete examples are in Appendix K. Following the format-correct reranking only, the enhancement is shown in Table 3, which suggests LLMs have the potential but are not ready to be directly deployed for the solution.

## 4.4 EMPIRICAL ANALYSIS OF GEOMETRIC CONDITIONS

Here, we mainly present two experimental analysis of geometric conditions, i.e., (1) the weight of $\mathcal{L}_{structure}$ in the structure-dominated encoder for cone, as expressed in Eq. 3 to preserve the monotonicity in the hyperbolic space, and (2) the relationship between structure loss and context loss in $\mathcal{L}_{total}$ expressed in Eq.13. Extensive experiments are placed in the appendix:

- The analysis for investigating the role of sequential self-attention mechanism with graph neural networks message-passing mechanism in the context-dominated encoder is in Appendix I.2.
- The analysis of varying the number of sampled hops can be found in Appendix I.3.
- The analysis of the difference between Euclidean and Hyperbolic manners for the structure-dominated encoder is in Appendix I.4
- The analysis of the relationship between the hard negative sampling and random sampling is in Appendix I.5.

First, we conduct the ablation study of the hyperbolic embedding to show its indispensability. In Table 4, we can see that totally removing the structure-dominated encoder (i.e., weight = 0) usually induces the worst performance.

Second, we conduct another ablation study for the ancestor, descendant, and sibling encoding and investigated their relationships. According to Eq. 10, we have $\beta$ for all sampled neighbors in general, $\mu$ for sampled descendants only, $\lambda$ for sampled ancestors only, and $\xi$ for sampled siblings only. Taking the SemEval-Food dataset as an example, in Table 5, we can observe that with all sampled nodes considered together, i.e., weight = 1111, the optimal results are obtained, compared with any ablation.

Table 4: Role of $\mathcal{L}_{structure}$ in Performance of SS-MONO (w/o AD) on SemEval-Food dataset.

| Weight of $\mathcal{L}_{structure}$ | Total | | | Leaf | | | Non-leaf | | |
|---|---|---|---|---|---|---|---|---|---|
| | MR ↓ | MRR ↑ | R@10 ↑ | MR ↓ | MRR ↑ | R@10 ↑ | MR ↓ | MRR ↑ | R@10 ↑ |
| 0 | 350.737 | 0.399 | 0.259 | **190.833** | 0.551 | 0.426 | 1219.779 | 0.074 | 0.024 |
| 0.1 | 349.031 | 0.399 | 0.315 | 230.161 | 0.670 | 0.607 | 933.872 | 0.190 | 0.091 |
| 0.3 | **304.774** | 0.428 | 0.322 | 222.891 | **0.698** | **0.657** | **727.834** | 0.215 | 0.057 |
| 0.5 | 315.792 | **0.430** | **0.338** | 228.177 | 0.690 | 0.642 | 768.466 | **0.225** | **0.098** |
| 0.7 | 389.381 | 0.358 | 0.270 | 279.644 | 0.626 | 0.533 | 956.356 | 0.146 | 0.063 |
| 1 | 335.416 | 0.391 | 0.305 | 211.904 | 0.679 | 0.600 | 943.098 | 0.171 | 0.080 |

Table 5: Ablation study of weight combinations $(\beta, \mu, \lambda, \xi)$ for objective function in $\mathcal{L}_{structure}$.

| Weight$(\beta, \mu, \lambda, \xi)$ | Total | | | | | | | | Leaf | | | Non-leaf | | |
|---|---|---|---|---|---|---|---|---|---|---|---|---|---|---|
| | MR ↓ | MRR ↑ | R@1 ↑ | R@5 ↑ | R@10 ↑ | P@1 ↑ | P@5 ↑ | P@10 ↑ | MR ↓ | MRR ↑ | R@10 ↑ | MR ↓ | MRR ↑ | R@10 ↑ |
| 1111 | **315.79** | **0.430** | **0.161** | **0.283** | **0.338** | **0.338** | **0.119** | **0.071** | 228.18 | **0.690** | **0.642** | **768.47** | **0.225** | **0.098** |
| 1110 | 323.55 | 0.217 | 0.039 | 0.093 | 0.129 | 0.081 | 0.039 | 0.027 | **132.21** | 0.445 | 0.289 | 1264.94 | 0.041 | 0.006 |
| 1001 | 430.85 | 0.379 | 0.148 | 0.254 | 0.309 | 0.311 | 0.107 | 0.065 | 264.74 | 0.665 | 0.621 | 1289.10 | 0.145 | 0.053 |
| 1000 | 345.07 | 0.272 | 0.068 | 0.164 | 0.199 | 0.142 | 0.069 | 0.042 | 205.83 | 0.449 | 0.343 | 1236.23 | 0.036 | 0.008 |
| 0111 | 1509.88 | 0.050 | 0.006 | 0.019 | 0.029 | 0.014 | 0.008 | 0.006 | 1373.50 | 0.078 | 0.059 | 2180.90 | 0.028 | 0.006 |
| 0000 | 1063.16 | 0.065 | 0.000 | 0.013 | 0.019 | 0.000 | 0.005 | 0.004 | 509.49 | 0.134 | 0.044 | 3787.18 | 0.012 | 0.000 |

## 5 RELATED WORK

Understanding complex graph structures with deep learning models has covered many tasks (e.g., node classification (Fu et al., 2024), link prediction (Tieu et al., 2025), graph classification (Fu et al., 2022), graph generation (He et al., 2025), graph regression (Wang et al., 2025)) and multiple applications (e.g., fake news detection (Fu & He, 2021), time-series attack (Lin et al., 2024), claim verification (Fang et al., 2025), multi-modal summarization (Ning et al., 2025), ads recommendation (Zhang et al., 2025), data cleaning (Li et al., 2025a)). As a special (hierarchical, directed, and text-attributed) graph structure, taxonomy paves the way for information retrieval, knowledge organization and question answering (Shen & Han, 2022). Based on non-generative models, classic taxonomy expansion/completion methods like (Zhang et al., 2021; Wang et al., 2022; Xue et al., 2024) focus on identifying the most suitable parent position and then adding the new item as the corresponding child node. Recently, the surge of large language models has inspired the exploration of graph structures, due to its broad world knowledge and reasoning ability (Wei et al., 2026; Li et al., 2025b; Fu et al., 2025; Xu et al., 2024). To start, Xu et al. (2022) proposed prompt tuning BERT for finding the hypernym of an incoming query and converting the hypernym prediction as a generation task. Later, Zeng et al. (2024a) introduced Chain-of-Layer to iteratively prompt LLMs for inducing taxonomy structure from a small set of entities. Mishra et al. (2024) fine-tuned Low Rank Adapter with Proximal Policy Optimization (PPO) for generating the hypernym of a query. To explore the node contextual features along the taxonomy's structural semantics, we propose SS-MONO to integrate the structural semantics and with concept textual semantics with the affordable LLM usage.

## 6 CONCLUSION

In this paper, we explored the intersection of hyperbolic structures in LLM embeddings and the scientific taxonomy expansion problem. Our study revealed that while LLMs possess strong representational capacity, they fail to reliably support domain-specific taxonomy expansion. To bridge this gap, we proposed SS-MONO, a self-supervised framework that borrows knowledge from LLMs but distills it into SLMs through structure- and semantics-aware training. Empirical results confirm that SS-MONO delivers substantial gains over both frozen LLMs and specialized deep learning models, establishing SLMs as a practical and scalable alternative for taxonomy expansion.

## ACKNOWLEDGMENT

This work is supported by National Science Foundation under Award No. IIS-2117902. The views and conclusions are those of the authors and should not be interpreted as representing the official policies of the funding agencies or the government.

ETHICS STATEMENT

This work adheres to the ICLR Code of Ethics. Our study relies exclusively on publicly available datasets and does not involve human subjects, personally identifiable information, or sensitive data. The findings are intended for scientific purposes only and do not pose foreseeable risks of harmful application or misuse. No conflicts of interest or external sponsorships have influenced this work.

REPRODUCIBILITY STATEMENT

We have made significant efforts to ensure the reproducibility of our results. Detailed descriptions of the model architecture and training procedure are provided in Sections 3 and 4.1. Additional hyperparameters, a detailed introduction of baseline models, implementation details, and evaluation metrics steps are documented in Appendices E, F, and G. To further support reproducibility, we provide code, configuration files, and data processing scripts in the anonymous GitHub repository `https://github.com/LiriFang/SS-Mono`. Together, these resources enable others to reproduce our experiments and validate our findings.

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

## A APPENDIX CONTENTS

## B LIMITATION

While our method demonstrates strong performance and general applicability, several limitations should be acknowledged:

**Fixed Sampling Depth.** The strategy of uniformly sampling 3-hop neighbors may not adequately capture essential context in taxonomies requiring deeper hierarchical insights, nor avoid irrelevant noise in shallow hierarchies.

**Dependence on LLMs.** Our method explores the effectiveness of LLM augmentations, meaning inaccuracies or biases present in the LLMs can propagate into the taxonomy expansions.

**LLM Model Size.** Due to resource limitations, we did not explore LLMs with more than 10B parameters. Investigating the ranking and retrieval capabilities of larger LLMs, both with and without fine-tuning, presents an interesting direction for future research.

## C HYPERBOLIC TRANSFORMATION OPERATIONS

### C.1 MOBIUS ADDITION ON POINCARÉ BALL

$$u \oplus_c v = \frac{\left(1 + 2c\langle u, v\rangle + c\|v\|^2\right) u + \left(1 - c\|u\|^2\right) v}{1 + 2c\langle u, v\rangle + c^2 \|u\|^2 \|v\|^2}, \quad u, v \in \mathbb{D}_c^d := \{x : c\|x\|^2 < 1\}. \quad (14)$$

where $\mathbf{u}$ and $\mathbf{v} \in \mathbb{D}^d$ and we use $c = 1$ throughout.

## C.2 EXPONENTIAL MAP

Let $\mathbb{D}^d = \{x \in \mathbb{R}^d : \|x\| < 1\}$ be the d-dimensional Poincaré ball (curvature -1). Define the conformal factor $\lambda_x = \frac{2}{1-\|x\|^2}$. The exponential map at $x \in \mathbb{D}^d$, $\exp_x : T_x\mathbb{D}^d \to \mathbb{D}^d$, is

$$\exp_x(v) = x \oplus \left( \tanh\left( \frac{\lambda_x \|v\|}{2} \right) \frac{v}{\|v\|} \right), \quad v \neq 0, \tag{15}$$

and $\exp_x(0) = x$. Here $\oplus$ denotes Möbius addition on the Poincaré ball (see Appendix C.1), and we identify $T_x\mathbb{D}^d \simeq \mathbb{R}^d$.

## C.3 LOGARITHMIC MAP

The inverse operation (logarithmic map) $\log_x : \mathbb{D}^d \to T_x\mathbb{D}^d$ for $x \in \mathbb{D}^d$ is

$$\log_x(y) = \frac{2}{\lambda_x} \operatorname{arctanh}\left( \| -x \oplus y \| \right) \frac{-x \oplus y}{\| -x \oplus y \|}, \quad y \neq x, \tag{16}$$

and $log_x(x) = 0$. Here, $\oplus$ is the Mobius addition (details in Appendix C.1) and $\operatorname{arctanh}$ is inverse hyperbolic tangent.

## C.4 SPECIAL CASE AT THE ORIGIN.

Since $\lambda_0 = \frac{2}{1-\|0\|^2} = 2$ and $0 \oplus w = w$, the exponential and logarithmic maps at the origin reduce to

$$\exp_0(v) = \tanh(\|v\|) \frac{v}{\|v\|}, \quad v \neq 0, \qquad \exp_0(0) = 0,$$

$$\log_0(y) = \operatorname{artanh}(\|y\|) \frac{y}{\|y\|}, \quad y \neq 0, \qquad \log_0(0) = 0.$$

# D  CONE AND APERTURE

As stated in (Ganea et al., 2018a), define a cone width function $\phi(\cdot)$ so that the resulting cones satisfies the transitivity of partial order in an embedding space, then, in our hyperbolic setting, we have

$$\forall u, v \in \mathbb{D}^d \setminus \{0\}, v \in \mathfrak{S}_u^{\phi(u)} \implies \mathfrak{S}_v^{\phi(v)} \subset \mathfrak{S}_u^{\phi(u)}. \tag{17}$$

where $\mathfrak{S}_u^{\phi(u)}$ is the cone of a point $u$ with the width function $\phi(u)$. Moreover, the Poincaré entailment cone can be defined as

$$\mathfrak{S}_u^{\phi(u)} = \left\{ v \in \mathbb{D}^d \mid \Xi(u, v) \leq \phi(u) \right\}, \tag{18}$$

where the half-aperture is $\phi(u) = \arcsin\left( K \frac{1-\|u\|^2}{\|u\|} \right)$, $\|u\| \geq \varepsilon$, and $K$ is a hyperparameter. For any $u, v \in \mathbb{D}_d \setminus B_d(O, \varepsilon)$, $\Xi(u, v)$ is the angle between the half-lines $\overrightarrow{uv}$ and $\overrightarrow{0u}$, defined by

$$\Xi(u, v) := \pi - \angle Ouv. \tag{19}$$

This angle has the closed form

$$\Xi(u, v) = \arccos\left( \frac{\langle u, v \rangle (1 + \|u\|^2) - \|u\|^2 (1 + \|v\|^2)}{\|u\| \cdot \|u - v\| \sqrt{1 + \|u\|^2 \|v\|^2 - 2\langle u, v \rangle}} \right), \tag{20}$$

where $O$ denotes the origin.

## E    DETAILED INTRODUCTION OF NON-LLM BASELINES

- Bilinear Model (Sutskever et al., 2009). A relational model infers whether particular unobserved relations are likely to be true.
- Neural Tensor Network (Socher et al., 2013). An expressive neural tensor network suitable for reasoning over relationships between two entities.
- TaxoExpan (Shen et al., 2020). A taxonomy expansion model leverages graph neural networks for the egonet structure to learn node embeddings to expand.
- ARBORIST (Manzoor et al., 2020). A taxonomy expansion model considers the heterogeneous relations encoded in the taxonomy context by integrating the embedding distance with geometric distance as the dynamic margin loss.
- TMN (Zhang et al., 2021). A ranking-based taxonomy completion model uses the triplet matching network and defines taxonomy completion as a parent-child edge ranking task.
- QEN (Wang et al., 2022). A ranking-based taxonomy completion model extends TMN by adding siblings as additional signals.
- TaxBox (Xue et al., 2024): A taxonomy expansion method that leverages box containment and center closeness to design two specialized geometric scorers within the box embedding space.

## F    IMPLEMENTATION OF LLM BASELINES

Existing studies on LLMs for knowledge graph completion primarily focus on term prediction, where the model is given a sampled path from a knowledge graph and tasked with predicting the next node (Sun et al., 2024; Sehwag et al., 2024). However, to the best of our knowledge, no prior work has explored the application of LLMs (not as a foundation model) to taxonomy expansion in the context of query-position ranking.

To address this gap, we investigate the performance of LLMs in retrieving and reranking the top $k$ candidate positions, adapting the problem to a query-position ranking setting. Evaluating an LLM's ability to retrieve and rerank an extensive list of candidate positions is nontrivial due to the task's inherent complexity.

Following the document retrieval setup in (Lee et al., 2024), we construct a prompt that includes instructions, a list of taxonomy edges with corresponding indexes, and examples. However, incorporating all candidate taxonomy edges in the prompt exceeds the context length of DeepSeek-R1-8B, Gemma-2-9B, and Llama-3-8B, even for the smallest dataset, SemEval-Food. To address this limitation, we randomly sample 500 edges and instruct LLMs to retrieve the top 10, returning each edge index and rank in a defined format: $<$ edge_id $>$ $p$:parent_id–$c$:child_id $<$ rank $>$ xx. For GPT-4o mini, we conducted experiments on the SemEval-Food dataset using all 7,313 candidate edges as multi-message input. For the other two datasets, which contain a substantially larger number of candidate edges, we randomly sampled 500 edges following the procedure described above to ensure the feasibility and consistency of evaluation. For hyperparameter settings, we set the maximum number of generation tokens to 1000 and the temperature to 0.2. The detailed prompt template is shown in Block 1.

Listing 1: Query-Position Ranking Prompt Template for DeepSeek-R1

```
You will give the entire list of edges in an existing taxonomy. Please rerank the given candidate edges based
on the similarity of meaning to the query node. To be specific, the insertion means the parent term (i.e., the
 first term) of the edge is the hypernym of the query term, and the child term (i.e., the second term) of the
edge is the hyponym of the query term.
The most relevant edges should be rank 1, meaning the query term should be inserted between the two nodes of
the most relevant edges.
Each candidate edges is in the format of <edge_id>:<edge> where <edge_id> is the unique identifier of the edge
, <edge> is the edge in the format of <parent> -> <child>. If the child term is empty, it means the parent
term is the leaf node of the taxonomy.

Here is the total list of the existing edges in the taxonomy:
/*
<edge_id>: {edge_id} <edge>{parent_name} -> {child_name}<end-edge>
...
*/
```

```
Please return the top 10 candidate edges based on the relevance to the query term. The rank of the candidate
edges should be in the format of <edge_id><rank>,<edge_id><rank>,...

Query term: abdominal pain. Description of the query term: Abdominal pain is sensation of discomfort, distress
, or agony in the abdominal region.
Please rerank the provided candidate edges following the format: `[<edge_id>edge_id rank>1, <edge_id>edge_id<
rank>2, ...]`.

Reranked list of candidate edges:
[<edge_id>p:signsandsymptoms,digestive-c:abdomen,acute<rank>1,
<edge_id>p:pain-c:abdomen,acute<rank>2,
<edge_id>p:pain-c:acutepain<rank>3,
<edge_id>p:pain-c:chronicpain<rank>4,
<edge_id>p:signsandsymptoms,digestive-c:nausea<rank>5,
<edge_id>p:signsandsymptoms,digestive-c:vomiting<rank>6,
<edge_id>p:pain-c:<rank>7,
<edge_id>p:signsandsymptoms,digestive-c:<rank>8,
<edge_id>p:abdomen,acute-c:<rank>9,
<edge_id>p:abdomen-c:<rank>10]

Query term: {kwargs['query_term']}. Description of the query term: {kwargs['query_term_description']}
Please rerank the provided {kwargs['number_of_candidate_edges']} candidate edges following the format: `[<
edge_id>edge_id<rank>1, <edge_id>edge_id<rank>2, ...]`.
Reranked list candidate edges:
"""
```

# G EVALUATION METRICS

Following the same setting with (Zhang et al., 2021; Wang et al., 2022), we report the ranking-based evaluation metrics to measure the performance of SS-MONO with baseline models. We first sort all candidate positions based on the matching score $F(\mathbf{H}_q, \mathbf{R}_{\text{LLM}}, \mathbf{R}_a, \mathbf{R}_d, \mathbf{R}_s)$ as Eq. 9 and then return the ranks of the ground-truth positions in the sorted candidate position list for each query node. The evaluation metrics include *Mean Rank*, *Mean Reciprocal Rank*, *Recall@k*, and *Precision@k*. In addition, we compare the metrics by three categories, i.e., the leaf query nodes, the non-leaf query nodes, and the total query nodes (including both leaf and non-leaf query nodes).

- *Mean Rank (MR)* measures the macro average ranking of ground-truth positions among all candidate positions. The lower *Mean Rank* is, the higher the ranking of the ground-truth position is among candidate positions.
- *Mean Reciprocal Rank (MRR)* measures the macro average reciprocal rank of all ground-truth positions. Therefore, the higher the *MRR* is, the higher the ranking of the ground truth position is among all candidate positions.
- *Recall@k (R@k)* calculates the number of ground-truth positions in the top-k candidate positions, averaged by the total counts of ground-truth positions for all queries.
- *Precision@k (P@k)* calculates the number of ground-truth positions in the top-k candidate positions, averaged by the total number of queries times k.

# H COMPREHENSIVE CASE STUDY

To further explain the performance, we generate the concrete prediction examples generated by SS-MONO, SS-MONO(w/o AD), and QEN for the SemEval-Food dataset, as shown in Table 6.

For the leaf node insertion, SS-MONO correctly predicts the proper position at the top 1 for query concepts *"stinger"* and *"papaya juice"*. The actual position of query concept *"malmsey"*, i.e., *"fortified wine - pseudo leaf"* is predicted at the second rank. SS-MONO correctly predicts the proper position at the top 1 for query concepts *"stinger"*, and *"malmsey"*. However, the proper position is ranked third for *"papaya juice"*. Therefore, we further investigate the description provided in the dataset for *"malmsey"* and *"fortified wine"*. However, the raw input node description of *"malmsey"* does not imply or contain information related to alcohol by volume. With LLM-augmented candidate position description, SS-MONO captures information related to *"malmsey"* with *"fortified wine"*.

As for non-leaf insertion, compared with the baseline model QEN, SS-MONO achieves better rankings for query concepts containing multiple true insertion positions, e.g., *"milk"* has 60 ground truth

Table 6: Case Study: Top-10 Predicted Candidate Positions Generated by SS-Mono vs. SS-Mono (w/o AD) vs. QEN. "p:" indicates the hypernym concept of the query concept, and "c:" indicates the hyponym concept of the query concept. The ground truth rank for non-leaf insertion is the mean rank.

| SS-Mono | Leaf | | | Non-leaf | | |
|---|---|---|---|---|---|---|
| Ranking/Query Concept | stinger | papaya juice | malmsey | milk | sparkling wine | frozen dessert |
| 1 | **p:cocktail-c:pseudo leaf** | p:herb-c:pseudo leaf | **p:fortified wine-c:pseudo leaf** | **p:beverage-c:pseudo leaf** | p:red wine-c:pseudo leaf | p:cream-c:pseudo leaf |
| 2 | p:martini-c:pseudo leaf | p:fruit juice-c:pseudo leaf | p:burgundy-c:pseudo leaf | **p:dairy product-c:pseudo leaf** | p:fortified wine-c:pseudo leaf | p:concoction-c:pseudo leaf |
| 3 | p:whiskey-c:pseudo leaf | **p:juice-c:pseudo leaf** | p:table wine-c:pseudo leaf | **p:nutriment-c:pseudo leaf** | p:burgundy-c:pseudo leaf | **p:dessert-c:pseudo leaf** |
| 4 | p:daiquiri-c:pseudo leaf | p:coffee substitute-c:pseudo leaf | p:red wine-c:pseudo leaf | p:concentrate-c:pseudo leaf | p:whiskey-c:pseudo leaf | p:consomme-c:pseudo leaf |
| 5 | p:vermouth-c:pseudo leaf | p:syrup-c:pseudo leaf | p:whiskey-c:pseudo leaf | p:beverage-c:elixir | p:table wine-c:pseudo leaf | p:gelatin dessert-c:pseudo leaf |
| 6 | p:gin-c:pseudo leaf | p:vitamin a-c:pseudo leaf | p:cocktail-c:pseudo leaf | p:nutriment-c:water soluble vitamin | p:wine-c:pseudo leaf | p:curd-c:pseudo leaf |
| 7 | p:sour-c:pseudo leaf | p:soft drink-c:pseudo leaf | p:sherry-c:pseudo leaf | p:beverage-c:ale | p:sherry-c:pseudo leaf | p:bite-c:pseudo leaf |
| 8 | p:cocktail-c:strawberry daiquiri | p:tea-c:pseudo leaf | p:bordeaux-c:pseudo leaf | p:beverage-c:chicory | p:stout-c:pseudo leaf | p:meal-c:pseudo leaf |
| 9 | p:highball-c:pseudo leaf | p:garlic-c:pseudo leaf | p:rum-c:pseudo leaf | p:beverage-c:potion | p:ale-c:pseudo leaf | p:ready mix-c:pseudo leaf |
| 10 | p:cocktail-c:nada daiquiri | p:cola-c:pseudo leaf | p:orange liqueur-c:pseudo leaf | p:beverage-c:highball | p:bordeaux-c:pseudo leaf | p:yogurt-c:pseudo leaf |
| Ground Truth Rank | 1 | 3 | 1 | 300.933 | 160.5 | 72.077 |

| SS-Mono (w/o AD) | Leaf | | | Non-leaf | | |
|---|---|---|---|---|---|---|
| Ranking/Query Concept | stinger | papaya juice | malmsey | milk | sparkling wine | frozen dessert |
| 1 | **p:cocktail-c:pseudo leaf** | **p:juice-c:pseudo leaf** | p:table wine-c:pseudo leaf | **p:beverage-c:pseudo leaf** | p:red wine-c:pseudo leaf | **p:dessert-c:pseudo leaf** |
| 2 | p:hot toddy-c:pseudo leaf | p:fruit juice-c:pseudo leaf | **p:fortified wine-c:pseudo leaf** | **p:nutriment-c:pseudo leaf** | p:table wine-c:pseudo leaf | p:gelatin dessert-c:pseudo leaf |
| 3 | p:highball-c:pseudo leaf | p:drinking water-c:pseudo leaf | p:burgundy-c:pseudo leaf | p:beverage-c:must | p:burgundy-c:pseudo leaf | p:yogurt-c:pseudo leaf |
| 4 | p:gin-c:pseudo leaf | p:fruit drink-c:pseudo leaf | p:mulled wine-c:pseudo leaf | **p:beverage-c:semi skimmed milk** | p:fortified wine-c:pseudo leaf | p:hors d'oeuvre-c:pseudo leaf |
| 5 | p:martini-c:pseudo leaf | p:orange juice-c:pseudo leaf | p:sherry-c:pseudo leaf | **p:beverage-c:pasteurized milk** | p:mulled wine-c:pseudo leaf | p:ice cream-c:pseudo leaf |
| 6 | p:cocktail-c:daiquiri | p:herb-c:pseudo leaf | p:red wine-c:pseudo leaf | **p:beverage-c:yak's milk** | p:sherry-c:pseudo leaf | p:gelatin-c:pseudo leaf |
| 7 | p:cocktail-c:martini | p:coffee liqueur-c:pseudo leaf | p:bordeaux-c:pseudo leaf | **p:beverage-c:low fat milk** | p:bordeaux-c:pseudo leaf | p:pate-c:pseudo leaf |
| 8 | p:cocktail-c:nada daiquiri | p:coffee liqueur-c:pseudo leaf | p:ale-c:pseudo leaf | p:dairy product-c:pseudo leaf | p:ale-c:pseudo leaf | p:stuffing-c:pseudo leaf |
| 9 | p:cocktail-c:shrimp cocktail | p:soft drink-c:pseudo leaf | p:whiskey-c:pseudo leaf | **p:beverage-c:formula** | p:whiskey-c:pseudo leaf | p:ragout-c:pseudo leaf |
| 10 | p:cocktail-c:vodka martini | p:sage-c:pseudo leaf | p:fortified wine-c:sherry | **p:beverage-c:mother's milk** | p:red wine-c:beaujolais | p:patty-c:pseudo leaf |
| Ground Truth Rank | 1 | 1 | 2 | 161.967 | 1111.500 | 31.077 |

| QEN | Leaf | | | Non-leaf | | |
|---|---|---|---|---|---|---|
| Ranking/Query Concept | stinger | papaya juice | malmsey | milk | sparkling wine | frozen dessert |
| 1 | **p:cocktail-c:pseudo leaf** | p:fruit juice-c:pseudo leaf | p:liqueur-c:pseudo leaf | p:canned food-c:pseudo leaf | p:weissbier-c:pseudo leaf | **p:dessert-c:pseudo leaf** |
| 2 | p:ale-c:pseudo leaf | p:ready mix-c:pseudo leaf | p:weissbier-c:pseudo leaf | **p:dairy product-c:pseudo leaf** | p:red wine-c:pseudo leaf | p:cocktail-c:pseudo leaf |
| 3 | p:condiment-c:pseudo leaf | p:herb tea-c:pseudo leaf | p:sour-c:pseudo leaf | p:wheat flour-c:pseudo leaf | p:sour-c:pseudo leaf | p:starches-c:pseudo leaf |
| 4 | p:green tea-c:pseudo leaf | p:syrup-c:pseudo leaf | p:cinnamon-c:pseudo leaf | p:cream cheese-c:pseudo leaf | p:fortified wine-c:pseudo leaf | p:bite-c:pseudo leaf |
| 5 | p:butter-c:pseudo leaf | **p:juice-c:pseudo leaf** | **p:fortified wine-c:pseudo leaf** | p:dainty-c:pseudo leaf | p:burgundy-c:pseudo leaf | p:gelatin-c:pseudo leaf |
| 6 | p:conserve-c:pseudo leaf | p:fruit drink-c:pseudo leaf | p:red wine-c:pseudo leaf | p:cheddar-c:pseudo leaf | p:vermouth-c:pseudo leaf | p:ice-c:pseudo leaf |
| 7 | p:ice-c:pseudo leaf | p:wheat flour-c:pseudo leaf | p:coffee liqueur-c:pseudo leaf | p:mead-c:pseudo leaf | p:bordeaux-c:pseudo leaf | p:green tea-c:pseudo leaf |
| 8 | p:ice cream-c:pseudo leaf | p:mead-c:pseudo leaf | p:vermouth-c:pseudo leaf | p:feed-c:pseudo leaf | p:candy-c:pseudo leaf | p:dark bread-c:pseudo leaf |
| 9 | p:spread-c:pseudo leaf | p:curd-c:pseudo leaf | p:bordeaux-c:pseudo leaf | p:ready mix-c:pseudo leaf | p:liqueur-c:pseudo leaf | p:margarine-c:pseudo leaf |
| 10 | p:gelatin dessert-c:pseudo leaf | p:pepper-c:pseudo leaf | p:burgundy-c:pseudo leaf | p:herb tea-c:pseudo leaf | p:cinnamon-c:pseudo leaf | p:ale-c:pseudo leaf |
| Ground Truth Rank | 1 | 5 | 5 | 920.500 | 277.000 | 94.769 |

insertion positions and *"frozen dessert"* has 13 ground truth positions. However, SS-Mono doesn't perform better than SS-Mono (w/o AD), but better than QEN.

# I    More Ablation Study and Hyperparameter Analysis

## I.1    Number of Candidate Edges for the Input of DeepSeek-R1-8B

The implementation (including prompt template and hyperparameter settings) is given in Appendix F. In short, we prompt the LLM with a pool of randomly sampled candidate parent–child edges, then have it retrieve the top 10 and rank them in descending order. To assess how the quantity of candidate positions affects LLM ranking performance, we conduct an ablation study by varying the number of candidate edges per query concept, ranging from 100, 500, and 900.

Overall, the results in Table 7 show a clear non-monotonic trend. Increasing the pool size from 100 to 500 yields the strongest gains. R@1 improves from 0.006 to 0.016 and P@1 from 0.014 to 0.033, indicating that many true parents of SemEval-Food concepts are recovered only when the LLM is allowed to propose a moderately larger set of edges. However, further increasing to 900 candidates does not uniformly improve top-k accuracy: while R@5/R@10 and P@5/P@10 continue to rise slightly (0.023 and 0.009/0.005), R@1 and P@1 degrade compared to the 500-candidate setting. This suggests that excessively large candidate sets dilute the signal with noisy or spurious edges, making it harder for the re-ranking module to surface the correct parent in the very top positions. Interestingly, for non-leaf nodes, the 900-candidate setting shows a noticeable improvement in R@10 (0.023), implying that deeper or more structurally ambiguous nodes benefit from a larger and more diverse candidate pool. These observations confirm that moderate candidate expansion provides the best balance between coverage and noise, whereas overly large LLM-generated pools introduce diminishing or even negative returns in precision-oriented metrics.

## I.2    Self-Attention in Context-Dominated Encoder

To demonstrate the effectiveness of the self-attention mechanism (SAM) employed in the Context-Dominated Encoder, we conducted comprehensive experiments evaluating three different aspects: (1) a baseline model without SAM (denoted by SS-Mono(w/o SAM)); (2) an ablation study replacing SAM with standard graph neural networks, specifically GAT and GCN; and (3) an extended ablation

Table 7: Performance of DeepSeek-R1-8B with different numbers of candidate edges on SemEval-Food dataset.

| Method | Total | | | | | | | | Leaf | | | Non-leaf | | |
|---|---|---|---|---|---|---|---|---|---|---|---|---|---|---|
| | MR ↓ | MRR ↑ | R@1 ↑ | R@5 ↑ | R@10 ↑ | P@1 ↑ | P@5 ↑ | P@10 ↑ | MR ↓ | MRR ↑ | R@10 ↑ | MR ↓ | MRR ↑ | R@10 ↑ |
| DeepSeek-R1-8B-100 | – | – | 0.006 | 0.006 | 0.006 | 0.014 | 0.003 | 0.001 | – | – | 0.007 | – | – | 0.006 |
| DeepSeek-R1-8B-500 | – | – | **0.016** | 0.016 | 0.016 | **0.033** | 0.007 | 0.003 | – | – | **0.028** | – | – | 0.005 |
| DeepSeek-R1-8B-900 | – | – | 0.010 | **0.023** | **0.023** | 0.020 | **0.009** | **0.005** | – | – | 0.022 | – | – | **0.023** |

study integrating structure loss into GNNs, resulting in GAT+Cone and GCN+Cone variants. As presented in Table 8, the proposed SS-MONO consistently outperforms these ablation models across almost all evaluated metrics, with the exceptions being the Mean Rank (MR) for Total and Leaf nodes. It may suggest that the sequence-based self-attention mechanism is typically more effective than various subgraph-based message passing mechanisms.

Table 8: Ablation results comparing SS-Mono and its Graph Neural Network variants on taxonomy expansion.

| Method | Total | | | | | | | | Leaf | | | Non-leaf | | |
|---|---|---|---|---|---|---|---|---|---|---|---|---|---|---|
| | MR ↓ | MRR ↑ | R@1 ↑ | R@5 ↑ | R@10 ↑ | P@1 ↑ | P@5 ↑ | P@10 ↑ | MR ↓ | MRR ↑ | R@10 ↑ | MR ↓ | MRR ↑ | R@10 ↑ |
| SS-Mono | 315.79 | **0.430** | **0.161** | **0.283** | **0.338** | **0.338** | **0.119** | **0.071** | 228.18 | **0.690** | **0.642** | 768.47 | **0.225** | **0.098** |
| SS-Mono (w/o SAM) | 1063.16 | 0.065 | 0.000 | 0.013 | 0.019 | 0.000 | 0.005 | 0.004 | 509.49 | 0.134 | 0.044 | 3787.18 | 0.012 | 0.000 |
| SS-Mono (GAT) | **578.83** | 0.215 | 0.003 | 0.071 | 0.145 | 0.007 | 0.030 | 0.030 | 249.77 | 0.429 | 0.293 | 2278.94 | 0.039 | 0.023 |
| SS-Mono (GAT + Cone) | 615.93 | 0.167 | 0.016 | 0.029 | 0.074 | 0.034 | 0.012 | 0.016 | 307.57 | 0.329 | 0.141 | 2133.04 | 0.042 | 0.023 |
| SS-Mono (GCN) | 928.45 | 0.113 | 0.000 | 0.035 | 0.048 | 0.000 | 0.015 | 0.010 | 150.25 | 0.235 | 0.095 | 4949.16 | 0.017 | 0.011 |
| SS-Mono (GCN + Cone) | 638.44 | 0.138 | 0.003 | 0.016 | 0.035 | 0.007 | 0.007 | 0.007 | **90.59** | 0.270 | 0.064 | 3469.03 | 0.030 | 0.012 |

## I.3 NUMBER OF HOPS

In the experiment, we show that the performance with sampling 3-hop neighbors for all datasets with a fair comparison, given the depth of (sub)trees is not deep, e.g., ranging from 1 to 8. Also, 3 is the fair depth to balance the useful information (near neighbors) and noise (far neighbors). To further justify the hyperparameter selection, we conducted experiments with 2-hop, 3-hop, and 4-hop sampling in Table 9. 3-hop sampling shows the most compelling performance across most of the performance metrics.

Table 9: Performance comparison across different hop sizes for evidence expansion.

| Method | Total | | | | | | | | Leaf | | | Non-leaf | | |
|---|---|---|---|---|---|---|---|---|---|---|---|---|---|---|
| | MR ↓ | MRR ↑ | R@1 ↑ | R@5 ↑ | R@10 ↑ | P@1 ↑ | P@5 ↑ | P@10 ↑ | MR ↓ | MRR ↑ | R@10 ↑ | MR ↓ | MRR ↑ | R@10 ↑ |
| 2hop | 321.49 | 0.383 | 0.145 | 0.238 | 0.289 | 0.304 | 0.100 | 0.061 | 255.07 | 0.637 | 0.567 | **682.51** | 0.172 | 0.059 |
| 3hop | 315.79 | **0.430** | **0.161** | **0.283** | **0.338** | **0.338** | **0.119** | **0.071** | 228.18 | **0.690** | **0.642** | 768.47 | **0.225** | **0.098** |
| 4hop | **305.02** | 0.406 | 0.141 | 0.244 | 0.293 | 0.297 | 0.103 | 0.061 | **217.74** | 0.617 | 0.546 | 779.33 | 0.232 | 0.082 |

## I.4 EUCLIDEAN VS. HYPERBOLIC ENCODER IN STRUCTURE-DOMINATED ENCODER

To more thoroughly assess the contribution of the hyperbolic encoder and nested entailment cone objective, we conduct an ablation in which the structural encoder is replaced with a purely Euclidean transformer architecture and the cone loss is substituted with a Euclidean ordering-constraint objective:

$$L = \tfrac{1}{2}\Big[\max\big(0,\, d(q,p) - d(p,c) + m\big) + \max\big(0,\, d(q,c) - d(p,c) + m\big)\Big],$$

where $d(\cdot)$ denotes the standard L2 distance. In this variant, we keep the same two-layer transformer encoder used in SS-MONO but remove all hyperbolic components, including the Euclidean-to-hyperbolic projection and the nested entailment cone mechanism. The resulting representations are fed directly into the Euclidean ordering loss, ensuring that the comparison isolates the geometric modeling choice rather than architectural differences.

Across all three datasets, as shown in Table 10, SS-MONO with the hyperbolic encoder consistently outperforms the Euclidean variant on every metric in the Total, Leaf, and Non-leaf evaluations,

underscoring the importance of hyperbolic representations and the cone-based objective for capturing hierarchical structure and achieving strong performance.

Table 10: Performance of SS-Mono with Hyperbolic and Euclidean Encoders on SemEval-Food, WordNet-Verb, and MeSH Datasets.

| Method | Total | | | | | | | | Leaf | | | Non-leaf | | |
|---|---|---|---|---|---|---|---|---|---|---|---|---|---|---|
| | MR↓ | MRR↑ | R@1↑ | R@5↑ | R@10↑ | P@1↑ | P@5↑ | P@10↑ | MR↓ | MRR↑ | R@10↑ | MR↓ | MRR↑ | R@10↑ |
| **SemEval-Food** | | | | | | | | | | | | | | |
| SS-Mono (Hyperbolic) | **239.169** | **0.400** | **0.186** | **0.299** | **0.325** | **0.392** | **0.126** | **0.068** | **143.937** | **0.705** | **0.645** | **756.735** | **0.147** | 0.059 |
| SS-Mono (Euclidean) | 654.712 | 0.175 | 0.019 | 0.077 | 0.100 | 0.041 | 0.032 | 0.021 | 263.317 | 0.298 | 0.148 | 2580.372 | 0.080 | **0.062** |
| **WordNet-Verb** | | | | | | | | | | | | | | |
| SS-Mono (Hyperbolic) | **1626.522** | **0.334** | **0.106** | **0.208** | **0.260** | **0.163** | **0.064** | **0.040** | **922.541** | **0.521** | **0.457** | **4551.311** | **0.122** | **0.035** |
| SS-Mono (Euclidean) | 3682.875 | 0.059 | 0.005 | 0.018 | 0.025 | 0.007 | 0.006 | 0.004 | 1362.167 | 0.074 | 0.021 | 13641.050 | 0.042 | 0.030 |
| **MeSH** | | | | | | | | | | | | | | |
| SS-Mono (Hyperbolic) | **436.820** | **0.427** | **0.074** | **0.197** | **0.288** | **0.173** | **0.093** | **0.068** | **390.717** | **0.570** | **0.476** | **540.551** | **0.334** | **0.166** |
| SS-Mono (Euclidean) | 8976.253 | 0.075 | 0.010 | 0.046 | 0.061 | 0.023 | 0.021 | 0.014 | 7038.222 | 0.039 | 0.026 | 13490.450 | 0.104 | 0.089 |

## I.5 HARD NEGATIVE SAMPLING AND RANDOM SAMPLING

To evaluate whether our negative-sampling strategy introduces bias or overestimates robustness, we designed a set of ablations that systematically vary the difficulty of negative examples. This analysis examines how different proportions of structurally local ("hard") and globally sampled ("random") negatives affect model behavior.

We consider two types of negatives:

- **Hard negatives** are constructed from the ego-network surrounding each candidate parent–child position. For a given query node, we extract egonets around both the true parent–child edges and the alternative candidate positions, and collect all edges within these neighborhoods that are neither gold positions nor edges directly incident to the query. This produces a pool of structurally plausible but incorrect placements that represent challenging local confounders.

- **Random negatives** are sampled uniformly from the global candidate pool of non-positive positions. These negatives represent structurally distant or unambiguous alternatives and provide broad coverage of the negative space.

We evaluate two sampling regimes that span a spectrum from structurally unbiased to highly local. We form mixed batches in which a proportion of negatives are hard negatives and the remainder are random negatives. We vary the hard-negative ratio across {0%, 10%, 30%, 50%, 70%, 90%}, keeping all other hyperparameters and batch construction identical across settings. This setup directly tests whether increasing structural locality in the negative pool leads to artificially inflated performance.

The ablation results (in Table 11) demonstrate a non-monotonic relationship between the proportion of hard negatives and overall performance. On SemEval-Food, for example, moderate ratios (10–30%) provide consistent improvements, while higher ratios (≥50%) lead to performance degradation, particularly for non-leaf concepts. Importantly, the random-only configuration remains a strong and stable baseline, indicating that the model does not depend heavily on structurally localized negatives to perform competitively.

Table 11: Effect of Hard Negative Sampling Ratio on SemEval-Food Dataset. Taking 0% ratio as the baseline, red means gain, blue means loss, and heat means degree.

| Hard Negative Samples | Total | | | | | | | | Leaf | | | Non-leaf | | |
|---|---|---|---|---|---|---|---|---|---|---|---|---|---|---|
| | MR↓ | MRR↑ | R@1↑ | R@5↑ | R@10↑ | P@1↑ | P@5↑ | P@10↑ | MR↓ | MRR↑ | R@10↑ | MR↓ | MRR↑ | R@10↑ |
| 0% | 239.169 | 0.400 | 0.186 | 0.299 | 0.325 | 0.392 | 0.126 | 0.068 | 143.937 | 0.705 | 0.645 | 756.735 | 0.147 | 0.059 |
| 10% | 233.394 | 0.391 | 0.199 | 0.322 | 0.334 | 0.419 | 0.135 | 0.070 | 108.495 | 0.749 | 0.696 | 847.900 | 0.117 | 0.057 |
| 30% | 268.193 | 0.364 | 0.158 | 0.283 | 0.312 | 0.331 | 0.119 | 0.066 | 107.294 | 0.717 | 0.644 | 1059.815 | 0.093 | 0.057 |
| 50% | 324.313 | 0.346 | 0.193 | 0.283 | 0.302 | 0.405 | 0.119 | 0.064 | 146.489 | 0.644 | 0.570 | 1290.743 | 0.073 | 0.056 |
| 70% | 354.153 | 0.330 | 0.141 | 0.251 | 0.289 | 0.297 | 0.105 | 0.061 | 114.738 | 0.583 | 0.516 | 1591.126 | 0.073 | 0.058 |
| 90% | 458.789 | 0.277 | 0.048 | 0.167 | 0.215 | 0.101 | 0.070 | 0.045 | 151.653 | 0.554 | 0.423 | 2045.659 | 0.060 | 0.052 |

Two factors contribute to the degradation observed at high hard-negative ratios:

**Overemphasis on highly ambiguous alternatives.** When most negatives originate from local egonets, the loss becomes dominated by extremely subtle or nearly indistinguishable negative examples. This encourages the model to overfit to fine-grained structural patterns specific to the training taxonomy rather than learning generalizable cues.

**Reduced coverage of the easy and medium negative space.** Random negatives help establish a broad decision margin by teaching the model what clearly cannot be a parent/child. Excessive reliance on hard negatives reduces exposure to this broader negative space, leading to mistakes on simple or moderately difficult negatives during evaluation.

## J  TEMPLATE FOR LLM ACHIEVING CALIBRATION

In this section, the prompting template for reranking calibration by LLMs is provided in Block 2. We deploy Llama3.1:8b (Touvron et al., 2023) for the calibration.

Listing 2: LLM Calibration Prompt Template

```
Please rerank the given candidate edges where a query term can be inserted. The insertion means the parent
term of the edge is the hypernym of the query term, and the child term is the hyponym of the query term.
Please rerank the given candidate edges based on the similarity of meaning to the query node. To be specific,
the insertion means the parent term (i.e., the first term) of the edge is the hypernym of the query term, and
the child term (i.e., the second term) of the edge is the hyponym of the query term. I will give you a rerank
task with ten candidate edges as an example to warm you up. After the warm-up, I will give you the arbitrary
number of queries and candidate edges. Please make sure the number of your reranks is consistent with the
number of the original candidate edges.

Here is one example reranking 10 candidate edges.
10 Candidate edges:
<edge_id>p:868-c:1106 <edge>milk -> raw milk<end-edge>
<edge_id>p:868-c:1253 <edge>milk -> sour milk <end-edge>
...
<edge_id>p:1061-c:644 <edge> porridge -> gruel <end-edge>

Query term: acidophilus milk. Description of the query term: acidophilus milk is milk fermented by bacteria;
used to treat gastrointestinal disorders.
Please rerank the provided candidate edges following the format: '[<edge_id>edge_id <rank>1, <edge_id>edge_id<
rank>2, ...]'.
Reranked list of candidate edges:
[<edge_id>p:868-c:<rank>1,
<edge_id>p:868-c:1253<rank>2,
<edge_id>p:868-c:513<rank>3,
...
<edge_id>p:321-c:1286<rank>10]

Here is the actual inputs:
{number_of_candidate_edges} Candidate edges:\n
{candidate_edges}
Query term: {query_term}.
Description of the query term: {query_term_description}
Please rerank the provided {number_of_candidate_edges} candidate edges following the format: '[<edge_id>
edge_id <rank>1, <edge_id>edge_id<rank>2 ...]'.
Reranked list candidate edges:
```

## K  FAILED EXAMPLE OF LLM CARIBRATION

### K.1  NO RANKING ANSWER

Two main cases are observed when no ranking answers are successfully parsed since LLMs refuse to generate edge indices provided in the input prompt. One case is LLM generating ideas or thoughts on how to solve the reranking task. The other case is LLM generating python scripts trying to solve the reranking task. We will list one example output with long thoughts and Python scripts in Block 3. As a result, LLMs generation can hardly follow the format requirement in the Block 2.

Listing 3: LLM Calibration Failed Case: No Ranking Answer, i.e., Generating Python Scripts

```
To solve this problem, I will use a combination of natural language processing (NLP) and machine learning
techniques to rerank the given candidate edges based on their similarity to the query term \"julep\".

Here's my approach:
```

```
1. **Preprocessing**: I'll first preprocess the text data by tokenizing the words in each edge and converting
them to lowercase.
2. **Feature extraction**: I'll extract features from each edge that are relevant to the query term \"julep\".
 These features might include:\n\t* Presence of specific words like \"bourbon\", \"sugar\", \"mint\", or \"
crushed ice\" in the edge.\n\t* Syntactic relationships between words, such as \"and\" or \"over\".
3. **Model training**: I'll train a machine learning model on a labeled dataset to learn the patterns and
relationships that are indicative of edges related to \"julep\". The model will be trained to predict the
relevance of each edge to the query term.\n4. **Reranking**: Once the model is trained, I'll use it to rerank
the given candidate edges based on their predicted relevance to the query term \"julep\".

Here's my implementation in Python:
```python
import re
from nltk.tokenize import word_tokenize
from sklearn.feature_extraction.text import TfidfVectorizer
from sklearn.model_selection import train_test_split
from sklearn.linear_model import LogisticRegression
from sklearn.metrics import accuracy_score

# Preprocess the text data
def preprocess_text(text):
    tokens = word_tokenize(text.lower())
    return ' '.join(tokens)

# Extract features from each edge
def extract_features(edge):
    features = []
    if 'bourbon' in edge or 'whiskey' in edge:
        features.append(1)
    # Presence of bourbon/whiskey
    ...
```

## K.2 HALLUCINATED EDGES IN THE EXISTING TAXONOMY

One example of hallucinated edges generated by LLMs is shown in Table 12. The LLM ReRanked List contains one hallucinated edge "p:620-c:375" by connecting two existing terms "p:620" from edge "p:620-c:" and "c:375" from edge "p:1247-c:375".

Table 12: LLM Calibration Case: Hallucinated Edges. The hallucinated edge example is **bolded**.

| Input List | p:1247-c: | p:1284-c: | p:540-c: | p:570-c: | p:359-c: | *p:620-c:* | p:986-c: | p:1337-c: | p:272-c: | p:1247-*c:375* |
|---|---|---|---|---|---|---|---|---|---|---|
| **LLM ReRanked List** | p:620-c: | p:540-c: | p:570-c: | **p:620-c:375** | p:986-c: | p:1337-c: | p:272-c: | p:359-c: | p:1247-c: | p:1284-c: |

## K.3 SHORTEN RANKING ANSWER

Some cases show that LLMs would shorten the reranked list by ranking several top edges and mentioning that all other edges are irrelevant. One example asking LLM to rerank 50 candidate positions is shown in Block 4.

Listing 4: LLM Calibration Case: The Shorten Ranking Answer

```
To rerank the candidate edges for the query term \"cinnamon toast\", I will analyze each edge and calculate a
similarity score based on the description of the query term. The scores are subjective and may vary depending
on individual perspectives.

Here's my analysis:
1. **Edge**: p:1364-c:966 <edge>toast -> orange toast<end-edge>\n\t* Score: 0 (orange toast is not related to
cinnamon)
2. **Edge**: p:1364-c: <edge>toast -> <end-edge>\n\t* Score: 5 (basic bread, but no flavor or topping
mentioned)
3. **Edge**: p:1302-c: <edge>stuffing -> <end-edge>\n\t* Score: 0 (stuffing is not related to toast or
cinnamon)
4. **Edge**: p:1096-c: <edge>quick bread -> <end-edge>\n\t* Score: 5 (quick bread is a type of bread, but no
flavor or topping mentioned)
...
20. **Edge**: p:1096-c:142 <edge>quick bread -> bran muffin<end-edge> (Score: 0)
21. **Edge**: p:412-c:1236 <edge>cracker -> soda cracker<end-edge> (Score: 5)
22. **Edge**: p:1096-c:737 <edge>quick bread -> johnnycake<end-edge> (Score: 0)

The top 3 edges with the highest scores are:
1. **Edge**: p:412-c:971 <edge>cracker -> oyster cracker<end-edge> (Score: 5)
2. **Edge**: p:331-c: <edge>clarified butter -> <end-edge> (Score: 5)
```

```
3. **Edge**: p:412-c:1236 <edge>cracker -> soda cracker<end-edge> (Score: 5)

These edges are the most relevant to the topic of toast with cinnamon, as they involve types of bread or
toppings that could be used in a similar context.

Please note that this ranking is based on a subjective scoring system and may not reflect the actual relevance
 or importance of each edge in a real-world scenario.
```

## L  IMPLEMENTATION DETAILS OF SS-MONO

### L.1  TAXONOMY EXPANSION VIA QUERY-POSITION MATCHING

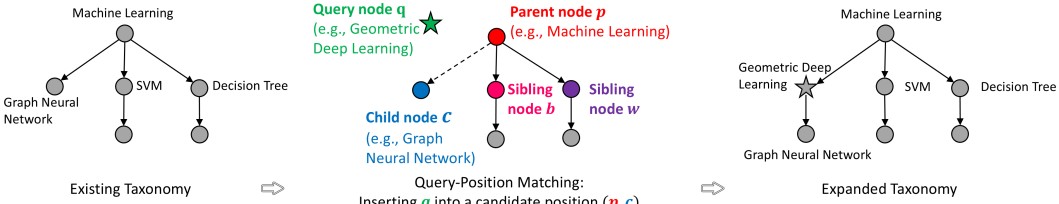

Figure 3: **Taxonomy Expansion Task** via **Query-Position Matching**. If query $q$ finds the best-matched position to insert, e.g., $(p, c)$, then it will break the existing edge $(p, c)$ and establish new edges $(p, q)$ and $(q, c)$.

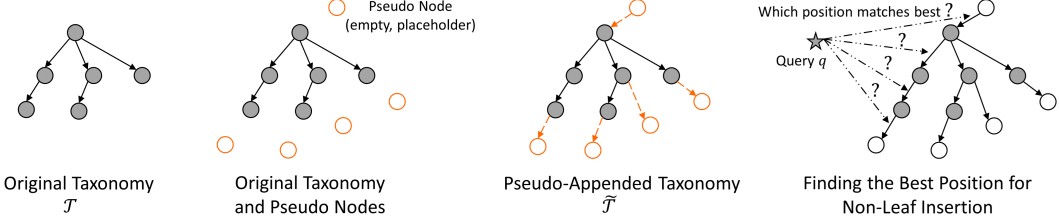

Figure 4: Establishing Pseudo-Appended Taxonomy $\tilde{\mathcal{T}}$ from $\mathcal{T}$ for Unifying Non-Leaf Insertion and Leaf Insertion.

Given an existing taxonomy $\mathcal{T} = (V, E)$ and a query node $q$ (i.e., a newly emerging concept that has not appeared in $\mathcal{T}$), the non-leaf taxonomy expansion task aims to expand the taxonomy $\mathcal{T}$ to the new taxonomy $\mathcal{T}_q$ by inserting the query node $q$ appropriately. To be specific, the query node $q$ seeks to match the best candidate position, i.e., an edge like $(p, c)$ in $\mathcal{T}$, and then adds new edges $(p, q)$ and $(q, c)$ by breaking the original edge $(p, c)$. For illustration, a non-leaf insertion example for expanding the existing taxonomy is illustrated in Figure 3.

During the non-leaf expansion task, to maintain the possibility of appending the query node as the leaf node, in (Zhang et al., 2021), authors propose to append *pseudo nodes* to the existing taxonomy $\mathcal{T}$ and make it a *pseudo-append taxonomy* $\tilde{\mathcal{T}}$. The pseudo nodes are empty placeholders with zero feature vectors. In this way, inserting leaf and non-leaf nodes into the existing taxonomy $\mathcal{T}$ can be unified by only inserting non-leaf nodes into the pseudo-append taxonomy $\tilde{\mathcal{T}}$. The corresponding procedures are illustrated in Figure 4.

### L.2  ENCODE CONTEXTUALIZED EMBEDDINGS FROM LANGUAGE MODELS

We use DistilBERT-base-uncased (Sanh et al., 2019) as the backbone pre-trained language model (PLM) to encode the input concept description sentence. Here, we describe the steps to obtain the node feature embedding $\mathbf{H} \in \mathbb{R}^{|V \cup \{q\}| \times h}$ from the input concept description sentence $\mathbf{X}$. The first step is to feed the description sentence to the backbone PLM, $\mathbf{Z} = \text{PLM}(X)$, where $\mathbf{Z} \in \mathbb{R}^{|V \cup \{q\}| \times L \times h}$ and $L$ is the maximum length of tokens in each description sentence. Then, an attention-pooling layer is adapted to pool the $\mathbf{Z}$ to node-level embedding $\mathbf{H}$.

$$\mathbf{H} = \text{softmax}(\mathbf{Z}\mathbf{W}_5)^T \mathbf{Z} \tag{21}$$

where $\mathbf{W}_5 \in \mathbb{R}^{m \times h}$ is the trainable parameter and $m$ is the dimension size to which the $L$ length tokens is compressed. $\mathbf{H} \in \mathbb{R}^{|V \cup \{q\}| \times m \times h}$. When $m = 1$, we can get $\mathbf{H} \in \mathbb{R}^{|V \cup \{q\}| \times h}$ after squeezing.

### L.3 NEURAL ARCHITECTURE AND HYPERPARAMETERS

The 2-layer transformer encoder is used for SAM with the number of attention heads as 8. The hidden dimension of the SAM layer is 256. The dimension project from SS-MONO is trained by a RiemannianAdam optimizer using a cosine learning rate scheduler. The learning rate is linearly warmed up from 0 to $5 \times 10^{-5}$ in the first 10% training steps. The margin $\gamma$ is set as 0.1. The initialization of curvature is set as 1 and is set as a trainable parameter. The numbers of training epochs for SemEval-Food, WordNet-Verb, and MeSH are 50, 40, and 40.

### L.4 REPRODUCIBILITY

The experiments are executed on a Tesla V100 (32GB) GPU machine. The implementation code can be found in `https://github.com/LiriFang/SS-Mono`.

## M EDGE DESCRIPTION GENERATION AND SANITY CHECKING

In this section, we demonstrate the procedure for generating edge descriptions and performing a sanity check.

### M.1 EDGE DESCRIPTION GENERATION

To provide relational context beyond standalone node definitions, we generate edge descriptions that aim to explicitly articulate the relationship between a hypernym and its hyponym. These descriptions serve as lightweight relational signals that the structural encoder can consume. To reduce hallucinations, we use a tightly constrained prompt that requires all statements to be grounded in the provided definitions and context. We deploy Gemma2:9b (Mesnard et al., 2024) for description generation. The prompt template can be found in Listing 5.

Listing 5: Description Generation Prompt Template

```
Known parents of {parent_name}: {parent_context}
Child term: {child_name}
Child definition: {child_definition}
Known children of {child_name}: {child_context}

Task: Write a concise relationship explanation (2--4 sentences) describing how the child relates to the parent
.
Ground every statement in the provided definitions/context and avoid inventing unsupported facts.
Mention at least one concrete trait that ties the child specifically to the parent category.
```

Given the prompt, one of the generated edge descriptions in the food taxonomy is as follows.
**Parent (445):** *dairy product* — "dairy product is milk and butter and cheese."
**Child (474):** *double creme* — "double creme is cream with a fat content of 48% or more."
**Generated description:**

> "Double creme is a type of dairy product because it is made from cream, which falls under the broader category of dairy products as defined by its inclusion in 'milk and butter and cheese.' Specifically, double creme's high fat content (48% or more) distinguishes it within the range of dairy products."

### M.2 SANITY CHECKING OF GENERATED DESCRIPTIONS

We validate the quality and faithfulness of generated edge descriptions using a two-stage protocol: (i) token-level lexical similarity and (ii) semantic evaluation via LLM-as-Judge.

- The first one is for measuring from the token level for the capability of LLM to maintain old knowledge and bring new knowledge, which is efficient for scaled datasets.

- For the plausible cases according to the first manner, LLM-as-Judge is leveraged to deeply analyze them by calling API.

Across both lexical and semantic evaluations, LLM-generated edge descriptions are largely faithful, semantically aligned, and reliable as relational augmentation. Even among the lowest-ROUGE cases, only a small minority exhibit semantic drift, validating LLM-generated relationship descriptions as a practical and effective mechanism for transferring relational knowledge to smaller structural models. We extend as follows.

### M.2.1 TOKEN-LEVEL LEXICAL SIMILARITY

We compute ROUGE-1, ROUGE-L, Novelty Ratio, and TF-IDF similarity between each edge description and the original term definitions. These metrics provide a coarse measure of grounding versus new content introduced by the LLM.

- **ROUGE-1** and **ROUGE-L**: capturing unigram (ROUGH-1) and longest common (ROUGE-L) subsequence overlap between the given two terms' description and LLM generated edge description.
- **Novelty Ratio** (i.e., 100% - token overlap ratio): assessing how much content in the generated description is newly introduced versus grounded in the original definition.
- **TF-IDF similarity**: measuring global lexical similarity beyond direct token overlap.

Table 13 summarizes results across all three datasets. Novelty ratios typically exceed 50%, while ROUGE scores fall in the 20–40% range, suggesting that LLMs introduce meaningful relational content while remaining partially grounded.

Table 13: Lexical similarity evaluation of LLM-generated edge descriptions across three datasets.

| Metric | SemEval-Food | | WordNet-Verb | | MeSH | |
|---|---|---|---|---|---|---|
| | Mean | Std | Mean | Std | Mean | Std |
| ROUGE-1 | 0.3965 | 0.1091 | 0.3012 | 0.0814 | 0.4102 | 0.1008 |
| ROUGE-L | 0.2733 | 0.0865 | 0.2027 | 0.0624 | 0.2441 | 0.0662 |
| Novelty Ratio | 0.6129 | 0.1352 | 0.7169 | 0.1000 | 0.5286 | 0.1444 |
| TF-IDF Similarity | 0.6081 | 0.1412 | 0.5894 | 0.1703 | 0.6034 | 0.1305 |

### M.2.2 LLM-AS-JUDGE SEMANTIC CONSISTENCY EVALUATION

Lexical metrics do not capture semantic correctness. To measure semantic alignment, we use GPT-4o as an LLM-as-Judge. To manage API cost, we evaluate only the 80 lowest-ROUGE edges per dataset since those are most likely to exhibit errors. Each edge receives three independent votes (Aligned / Partially Aligned / Misaligned) using the prompt template in Listing 6.

Listing 6: Description Sanity Check Prompt Template

```
You are a taxonomy quality reviewer. Given the canonical wiki-style node descriptions, decide whether the
provided edge description is factually correct, aligned with the parent/child terms, and free of hallucinated
claims.

Parent Term: {parent_term}
Parent Definition: {parent_definition}

Child Term: {child_term}
Child Definition: {child_definition}

Edge Description:
\"\"\"{edge_description}\"\"\"

Instructions:
1. Compare the edge description against the parent/child definitions. Note any hallucinated entities or
attributes that contradict the references.
2. Flag missing information that prevents you from concluding the relation.
3. If a parent/child definition itself appears off-topic or inconsistent with the taxonomy scope, record it
under reference_issues (do not change the verdict to compensate).
4. Respond with a short JSON object: {{"verdict": "<Aligned|Partial|Misaligned>", "issues": ["string"], "
missing_information": ["string"], "reference_issues": ["string"]}}.
```

```
5. References to placeholder pseudo nodes (e.g., "pseudo root" or "pseudo leaf") are acceptable and should not
   be flagged or treated as missing information solely for being placeholders or lacking extra detail.
```

Results are shown in Table 14. Despite low lexical overlap, only a small fraction ($\leq 10\%$) of descriptions are misaligned, demonstrating that low-ROUGE generations can still be semantically correct.

Table 14: LLM-as-Judge semantic consistency evaluation on the 80 lowest-ROUGE edge descriptions per dataset.

| Dataset | Aligned | Partial | Misaligned |
|---|---|---|---|
| SemEval-Food | 47/80 (58.75%) | 25/80 (31.25%) | 8/80 (10.00%) |
| WordNet-Verb | 36/80 (45.00%) | 40/80 (50.00%) | 4/80 ( 5.00%) |
| MeSH | 58/80 (72.50%) | 19/80 (23.75%) | 3/80 ( 3.75%) |

## N    FINE-TUNING LLM FOR TAXONOMY EXPANSION TASK

To test the adaptability of our pipeline to off-the-shelf language models, we adopt TinyLlama-1.1B-intermediate-step-1431k-3T (Zhang et al., 2024) as a representative lightweight LLM and attach a LoRA adapter (Hu et al., 2022) to the last four transformer layers (for computational efficiency). Concretely, we fine-tune the attention and feed-forward projection modules (`q_proj`, `k_proj`, `v_proj`, `o_proj`, `gate_proj`, `up_proj`, `down_proj`) while freezing the token embedding layer. The TinyLlama encoder is trained under the same learning objective as SS-Mono, using the loss in Eq. (14), ensuring a consistent optimization setup for comparison.

This experiment demonstrates that our geometric encoder is compatible with pretrained LLM checkpoints: integrating an LLM encoder requires only minor architectural changes and remains computationally lightweight. As shown in Table 15, the fine-tuned TinyLlama variant achieves comparable MR and MRR and notably improves leaf insertions, but it degrades performance on non-leaf nodes. These results suggest that our LLM-SLM-distillation design is not worse than (or even better than) LLM fine-tuning.

Table 15: Performance of SS-Mono with Fine-Tuned TinyLlama on SemEval-Food Dataset.

| Method | Total | | | | | | | | Leaf | | | Non-leaf | | |
|---|---|---|---|---|---|---|---|---|---|---|---|---|---|---|
| | MR↓ | MRR↑ | R@1↑ | R@5↑ | R@10↑ | P@1↑ | P@5↑ | P@10↑ | MR↓ | MRR↑ | R@10↑ | MR↓ | MRR↑ | R@10↑ |
| SS-Mono | **239.169** | **0.400** | **0.186** | **0.299** | **0.325** | **0.392** | **0.126** | **0.068** | 143.937 | 0.705 | 0.645 | **756.735** | **0.147** | **0.059** |
| TinyLlama (fine-tuned) | 253.911 | 0.373 | 0.122 | 0.241 | 0.309 | 0.257 | 0.101 | 0.065 | **73.851** | **0.754** | **0.652** | 1139.808 | 0.080 | 0.045 |

## O    LLM AUGMENTED DESCRIPTIONS WITH NON-LEAF EXPANSION

We analyze why LLM-Augmented Descriptions (AD) tend to improve leaf-node insertion more consistently than non-leaf insertion, as shown in Table 2. Our observations indicate that this difference arises from how LLM-generated descriptions interact with the structural roles of different node types in a taxonomy along three dimensions.

**Leaf-node descriptions are structurally consistent and aligned with the insertion task.**    For leaf nodes, the LLM-generated descriptions primarily emphasize the concept's global semantic identity, such as being a terminal category or a fine-grained subtype. Since leaf nodes do not have children, these descriptions remain concise, stable, and semantically homogeneous across examples. This consistency provides the scorer with clearer signals regarding the node's appropriate position in the taxonomy.

**Non-leaf descriptions exhibit higher diversity due to dual relational roles.**    Non-leaf nodes participate in both upward (parent-facing) and downward (child-facing) relationships. As a result, their descriptions must encode more heterogeneous and relationally complex information. Empirically,

LLMs produce a wider range of phrasings for non-leaf nodes because the prompts must articulate how multiple concepts relate rather than describe an isolated entity. This broader linguistic variability increases variance in the augmented text representations.

**Variability in AD interacts with fine-grained structural ranking and can dilute local signals.** Intermediate insertion is a fine-grained ranking problem in which the scorer must distinguish among many structurally similar alternatives. When ADs for non-leaf nodes exhibit high variance, the resulting representations provide weaker or less discriminative cues for resolving subtle local structural differences. In contrast, leaf-node insertion benefits from the more uniform and taxonomy-aligned descriptions.

To sum up, LLM Augmented Descriptions is, so far, a viable solution that provides additional knowledge for the existing taxonomy and can be leveraged to improve the SLM's performance based on our designed geometric deep learning constraints and self-supervised learning approach, as shown in the extensive experiments. Indeed, it is not perfect and has the latent drawback discussed above, and we are more than willing to make it our future research direction. A few possible directions are listed below:

- **Incorporating external web knowledge for richer edge descriptions.** Future extensions may integrate controlled web search so that the LLM can retrieve verifiable information about concept relationships. This has the potential to generate more accurate, edge-oriented descriptions, particularly for non-leaf nodes whose semantics depend on multiple relational contexts.

- **Leveraging local neighborhood structure during AD generation.** Conditioning description generation on a node's local subgraph—such as siblings, parents, children, or subtree summaries—can better ground the textual output in the underlying taxonomy. Such structure-aware prompting may reduce semantic drift and lower variance in AD for non-leaf nodes.

- **Developing multi-step, grounded generation pipelines.** A more robust AD pipeline can combine (i) local-structure grounding, (ii) retrieved external knowledge, and (iii) a self-critique or refinement step. This multi-stage procedure aims to stabilize the textual signal, filter inconsistent relational statements, and produce richer, higher-fidelity descriptions that support fine-grained non-leaf insertion.

## P  THE USE OF LLMS

As required by the ICLR 2026 Submission Policy, we made the following statement for the use of LLMs. We used large language models (LLMs) as controlled assistive tools for writing, specifically to check grammar and improve clarity. All outputs were reviewed and edited by the authors, who take full responsibility for the final content. Different LLMs were also included in experiments as part of evaluating their effectiveness for the taxonomy expansion task.

