# OpenReview forum: "Geometric Constraints for Small Language Models to Understand and Expand Scientific Taxonomies"
_ICLR.cc/2026/Conference — ICLR 2026 Poster_

### Official Review · Reviewer_APGH · 2025-10-30

**Soundness:** 2
**Presentation:** 2
**Contribution:** 2
**Rating:** 4
**Confidence:** 3

**Summary:**

This paper introduces SS-Mono, a pipeline targeting efficient and accurate scientific taxonomy expansion by transferring hierarchical knowledge from Large Language Models to Small Language Models. SS-Mono integrates a local taxonomy augmentation phase using LLMs, a self-supervised geometric fine-tuning process with hyperbolic constraints, and LLM-based calibration for candidate ranking. The approach is evaluated on several taxonomy expansion benchmarks (SemEval-Food, WordNet-Verb, and MeSH), showing that a fine-tuned SLM outperforms both frozen LLMs and domain-specific baselines for both leaf and non-leaf node insertions.

**Strengths:**

S1: The structure of SS-Mono is modular and well-motivated: it combines structure-dominated (hyperbolic metric) and context-dominated (semantic, LLM-augmented) encoders. This dual design is well illustrated in Figure 1, which shows how LLMs are used to enrich edge semantics and how structural relations are composed.

S2: Extensive experiments (see Tables 2 and 3) cover multiple datasets and include a wider range of both classical and recent baselines, including neural and LLM-based approaches. The results show consistent, substantial gains of SS-Mono, especially when augmented with LLM outputs, across MR, MRR, and Recall/Precision metrics.

S3: Self-supervised training enables the model to leverage existing taxonomies without costly annotation. The mathematical formulations (see Equations 1-11 and Cone Loss in Section 3.2) are clear, concise, and consistent with state-of-the-art hyperbolic learning approaches.

**Weaknesses:**

W1: While the paper claims that "LLMs have potential but are not ready to be directly deployed," the LLM calibration component appears to be more of an optional enhancement than a robust part of the pipeline. Figure 2 exposes high failure rates in LLM output for reranking (frequent hallucinated edges or incomplete lists), and Section 4.3 provides only a partial mitigation of this challenge. The improvements from LLM calibration, as shown in Table 3, are not always consistent or predictable.

W2: The model relies on self-supervised negative sampling and LLM-augmented context for candidate positions. The precise method for sampling hard negatives is tailored closely to the taxonomy structure as described in Section 3.4, but it is unclear whether the negative sampling strategy introduces bias or overestimates the model’s robustness to challenging/far-out queries.

W3: The pipeline discusses SLM fine-tuning with geometric constraints, but does not deeply analyze (or ablate) alternative LLM fine-tuning approaches (domain adaptation, prompt-tuning, etc.). This limits direct attribution of SLM’s advantages to the geometric approach, rather than simply to the smaller/faster architecture or training regime.

**Questions:**

Q1: Have the authors conducted any systematic analysis of the types or frequency of semantic drift or spurious insertions caused by LLM-augmented edge descriptions? If not, can they provide statistics or example cases showing where LLM guidance introduces incorrect or misleading hierarchy insertions?

Q2: Can the negative sampling strategy in self-supervised optimization be further detailed or ablated? Is there evidence that specific hard negative choices strongly affect generalization versus random negatives?

---

> ### Author Response · Authors · 2025-11-28
> **Rebuttal Answer from Authors (Part I)**
>
> Dear Reviewer APGH,
>
> Thanks very much for your review! We want to express our sincere gratitude to your appreciation of our paper’s motivation of introducing hyperbolic conditions into the semantic space, extensive experiments of broad classical and recent baselines, and also self-supervised training with the clear algorithmic illustration.
>
> Your suggestions are very actionable to further improve the paper’s quality. We seriously prepare the answer below. Moreover, we have updated the paper accordingly:
> - We added Appendix I.5 to systematically analyze the impact of negative sampling, which clearly discusses the type of different negative sampling, the source of them, and the varying ratio of them for the downstream task performance.
> - We added Appendix N to analyze the performance of directly fine-tuning LLMs with our LLM-to-SLM distillation design.
> - We added Appendix M to systematically analyze LLM-augmented edge description with various check manners.
>
> ---
>
> > W1. While the paper claims that "LLMs have potential but are not ready to be directly deployed," the LLM calibration component appears to be more of an optional enhancement than a robust part of the pipeline. Figure 2 exposes high failure rates in LLM output for reranking (frequent hallucinated edges or incomplete lists), and Section 4.3 provides only a partial mitigation of this challenge. The improvements from LLM calibration, as shown in Table 3, are not always consistent or predictable.
>
> We agree with your visionary insights! In short, our design is exactly echoing your suggestion.
>
> Moreover, in the paper, LLM calibration is not positioned as a core dependency of SS-Mono but rather as an **optional** post-hoc enhancer to explore the contribution of LLMs as a reranker.
>
> Our findings consistently reflect this intention.
>
> - In Section 4.3 and Figure 2 of the paper, we explicitly show that current LLMs (e.g., Llama-3.1-8B) often struggle with long-list reranking—failing with _hallucinated edges_, _format violations_, and _list truncation_. These observations reinforce our statement that LLMs are “not ready to be directly deployed,” and they justify why SS-Mono relies primarily on its SLM-based structure-semantic encoder.
> - Table 3 further validates this design choice: although certain settings (e.g., top-50 or top-100) yield modest improvements, others provide negligible or inconsistent gains. This variability supports our decision to treat calibration as a supplementary module rather than a central architectural component.
>
> In summary, the **calibration module is intentionally lightweight, optional, and conservatively applied**; its inconsistent benefit profiles directly reflect limitations of current LLMs rather than deficiencies of the main SS-Mono pipeline.
>
> > **W2** (and **Q2**). The model relies on self-supervised negative sampling and LLM-augmented context for candidate positions. The precise method for sampling hard negatives is tailored closely to the taxonomy structure as described in Section 3.4, but it is unclear whether the negative sampling strategy introduces bias or overestimates the model’s robustness to challenging/far-out queries.
>
> To evaluate whether our negative-sampling strategy introduces bias or overestimates robustness, we designed a set of ablations that systematically vary the difficulty of negative examples and **added them to Appendix I.5**. This analysis examines how different proportions of structurally local ('hard') and globally sampled ('random') negatives affect model behavior. Next, we extend the analysis from three aspects.
>
> ## 1. Construction of Negatives
>
> We consider two types of negatives:
> - **Hard negatives** are constructed from the ego-network surrounding each candidate parent--child position. For a given query node, we extract egonets around both the true parent--child edges and the alternative candidate positions, and collect all edges within these neighborhoods that are neither gold positions nor edges directly incident to the query. This produces a pool of structurally plausible but incorrect placements that represent challenging local confounders.
> - **Random negatives** are sampled uniformly from the global candidate pool of non-positive positions. These negatives represent structurally distant or unambiguous alternatives and provide broad coverage of the negative space.

---

> ### Author Response · Authors · 2025-11-28
> **Rebuttal Answer from Authors (Part II)**
>
> ## 2. Negative Sampling Regimes
>
> We evaluate two sampling regimes that span a spectrum from structurally unbiased to highly local. We form mixed batches in which a proportion of negatives are hard negatives and the remainder are random negatives. We vary the hard-negative ratio across \{0\%, 10\%, 30\%, 50\%, 70\%, 90\%\}, keeping all other hyperparameters and batch construction identical across settings. This setup directly tests whether increasing structural locality in the negative pool leads to artificially inflated performance.
>
> The ablation results in the below table demonstrate a non-monotonic relationship between the proportion of hard negatives and overall performance. On SemEval-Food, for example, moderate ratios (10--30\%) provide consistent improvements, while higher ratios ($\geq$50\%) lead to performance degradation, particularly for non-leaf concepts. Importantly, the random-only configuration remains a strong and stable baseline, indicating that the model does not depend heavily on structurally localized negatives to perform competitively.
>
> ## 3. Result Analysis
>
> Two factors contribute to the degradation observed at high hard-negative ratios:
> - **Overemphasis on highly ambiguous alternatives**. When most negatives originate from local egonets, the loss becomes dominated by extremely subtle or nearly indistinguishable negative examples. This encourages the model to overfit to fine-grained structural patterns specific to the training taxonomy rather than learning generalizable cues.
> - **Reduced coverage of the easy and medium negative space**. Random negatives help establish a broad decision margin by teaching the model what clearly cannot be a parent/child. Excessive reliance on hard negatives reduces exposure to this broader negative space, leading to mistakes on simple or moderately difficult negatives during evaluation.
>
> Table. Effect of Hard Negative Sampling Ratio on SemEval-Food Dataset.
> | Ratio of Hard Negative Samples |  Total  |       |       |       |        |       |       |        |   Leaf  |       |        | Non-leaf |       |        |
> |------------------------------------|:-------:|------:|------:|------:|-------:|------:|------:|-------:|:-------:|------:|-------:|:--------:|------:|-------:|
> |                                    |    MR ↓ | MRR ↑ | R@1 ↑ | R@5 ↑ | R@10 ↑ | P@1 ↑ | P@5 ↑ | P@10 ↑ |    MR ↓ | MRR ↑ | R@10 ↑ |     MR ↓ | MRR ↑ | R@10 ↑ |
> | 0\%                               | 239.169 | 0.400 | 0.186 | 0.299 |  0.325 | 0.392 | 0.126 |  0.068 | 143.937 | 0.705 |  0.645 |  756.735 | 0.147 |  0.059 |
> | 10\%                               | **233.394** | 0.391 | **0.199** | **0.322** |  **0.334** | **0.419** | **0.135** |  **0.070** | **108.495** | **0.749** |  **0.696** |  847.900 | 0.117 |  0.057 |
> | 30\%                               | 268.193 | 0.364 | 0.158 | 0.283 |  0.312 | 0.331 | 0.119 |  0.066 | 107.294 | 0.717 |  0.644 | 1059.815 | 0.093 |  0.057 |
> | 50\%                               | 324.313 | 0.346 | 0.193 | 0.283 |  0.302 | 0.405 | 0.119 |  0.064 | 146.489 | 0.644 |  0.570 | 1290.743 | 0.073 |  0.056 |
> | 70\%                               | 354.153 | 0.330 | 0.141 | 0.251 |  0.289 | 0.297 | 0.105 |  0.061 | 114.738 | 0.583 |  0.516 | 1591.126 | 0.073 |  0.058 |
> | 90\%                               | 458.789 | 0.277 | 0.048 | 0.167 |  0.215 | 0.101 | 0.070 |  0.045 | 151.653 | 0.554 |  0.423 | 2045.659 | 0.060 |  0.052 |
>
> ---
>
> > W3. The pipeline discusses SLM fine-tuning with geometric constraints, but does not deeply analyze (or ablate) alternative LLM fine-tuning approaches (domain adaptation, prompt-tuning, etc.). This limits direct attribution of SLM’s advantages to the geometric approach, rather than simply to the smaller/faster architecture or training regime.
>
> The suggested comparison, i.e., prompting LLMs to perform pairwise insertion evaluation followed by re-ranking, was considered in the original submission. However, in practice, LLMs struggle when asked to rank over the full candidate space: their recall@k and precision@k metrics are nearly zero, as shown in Table 2. Because the ground-truth parent positions almost never appear in the top-k list, a re-ranking step cannot provide meaningful calibration or improvement. As a result, such a baseline would not offer a fair or informative comparison.
>
> To deeply address your concern, we incorporate an LLM as the pretrained encoder and fine-tune it inside our pipeline. We report these results to demonstrate the role LLM representations can play when integrated more directly into our framework. **The below content is added into the paper Appendix N**.

---

> ### Author Response · Authors · 2025-11-28
> **Rebuttal Answer from Authors (Part III)**
>
> To test the adaptability of our pipeline, we adopt TinyLlama-1.1B-intermediate-step-1431k-3T [1] as a representative lightweight LLM and attach a LoRA adapter [2] to the last four transformer layers due to computational resource constraints.
>
> Specifically, we fine-tune the attention and feed-forward projection modules, including q_proj, k_proj, v_proj, o_proj, gate_proj, up_proj, and down_proj, while freezing the token embedding layer. TinyLlama-1.1B is fine-tuned under the same learning objective as the original SS-Mono, employing the loss function in Equation 14 to ensure a consistent optimization setup for comparison.
>
> This experiment further confirms that our geometric deep learning objective is easily compatible with off-the-shelf LLM checkpoints: only minimal modifications are needed to plug in an LLM encoder, and training remains computationally lightweight.
>
> In the below table, the results show comparable mean ranks (MR and MRR) across all testing nodes and better results for leaf node insertions; however, worse results for non-leaf node insertions. The empirical results show that LLM fine-tuning does not necessarily lead to consistent improvements.
>
> As shown below, the fine-tuned TinyLlama variant achieves comparable MR and MRR and notably improves leaf insertions, but it degrades performance on non-leaf nodes. These results suggest that our LLM-SLM-distillation design is not worse than (or even better than) LLM fine-tuning.
>
> Table. Performance of SS-Mono with Fine-Tuned LLM on SemEval-Food Dataset
> | Method            |  Total  |        |        |        |         |        |        |         |    Leaf   |         |         |  Non-leaf |         |         |
> |:-----------------:|:-------:|:------:|:------:|:------:|:-------:|:------:|:------:|:-------:|:---------:|:-------:|:-------:|:---------:|:-------:|:-------:|
> |                   |  MR(↓)  | MRR(↑) | R@1(↑) | R@5(↑) | R@10(↑) | P@1(↑) | P@5(↑) | P@10(↑) |   MR(↓)   |  MRR(↑) | R@10(↑) |   MR(↓)   |  MRR(↑) | R@10(↑) |
> | SS-Mono | **239.169** | **0.4**    | **0.186**  | **0.299**  | **0.325**   | **0.392**  | **0.126**  | **0.068**   | 143.93667 | 0.70478 | 0.64539 | **756.73471** | **0.14665** | **0.05882** |
> | Fine-Tuned TinyLlama    | 253.911 | 0.373  | 0.122  | 0.241  | 0.309   | 0.257  | 0.101  | 0.065   | **73.851**    | **0.754**   | **0.652**   | 1139.808  | 0.08    | 0.045   |
>
> Reference:
>
> [1] Zhang et al., Tinyllama: An open-source small language model, arXiv 2024
>
> [2] Hu et al., LoRA: Low-Rank Adaptation of Large Language Models, ICLR 2022
>
> ---
>
> > Q1. Have the authors conducted any systematic analysis of the types or frequency of semantic drift or spurious insertions caused by LLM-augmented edge descriptions? If not, can they provide statistics or example cases showing where LLM guidance introduces incorrect or misleading hierarchy insertions?
>
> We understand your concern. Therefore, we would like to give a holistic analysis:
> - **First**, we provide examples of our prompt template which was used to generate LLM-augmented edge description.
> - **Second**, we present the results of various sanity checks, which demonstrate that the LLM generated edge description is trustworthy. The below answer is now included in Appendix M.
>
> ## 1. Edge Description Generation Process and Example
> The edge description is designed to explicitly describe the relationship between a hypernym and its hyponym. This additional relational context enriches the information available to the model beyond term descriptions alone. To minimize the risk of hallucination or unsupported claims, we incorporated explicit constraints within the prompt instructions. The full prompt template for generating LLM-based edge descriptions is shown below.
> ```
> Known parents of {parent_name}: {parent_context}
> Child term: {child_name}
> Child definition: {child_definition},
> Known children of {child_name}: {child_context}",
>
> Task: Write a concise relationship explanation (2-4 sentences) describing how the child relates to the parent. Ground every statement in the provided definitions/context and avoid inventing unsupported facts. Mention at least one concrete trait that ties the child specifically to the parent category.
> ```
>
> ### 1.1 Example edge description generated by LLMs:
>
> Input:
> - Parent node, i.e., 445: dairy product: dairy product is milk and butter and cheese
> - Child node, i.e.,  474: double creme: double creme is cream with a fat content of 48% or more
>
> Output:
> - Edge Description: {"445-474": "Double creme is a type of dairy product because it is made from cream, which falls under the broader category of dairy products as defined by its inclusion in \"milk and butter and cheese.\"  \n\nSpecifically, double creme's high fat content (48% or more) distinguishes it within the range of dairy products."}

---

> > ### Author Response · Authors · 2025-11-28
> > **Rebuttal Answer from Authors (Part IV)**
> >
> > ## 2. Sanity Check
> >
> > To further assess the quality and faithfulness of the generated edge descriptions, we conducted **a systematic sanity check** comparing each generated description with its corresponding term description in the original taxonomy dataset.
> >
> > This validation step consists of two complementary analyses: **token-level lexical similarity** and **LLM-based semantic consistency evaluation (LLM-as-Judge)**.
> > - The first one is for measuring from the token level for the capability of LLM to maintain old knowledge and bring new knowledge, which is efficient for scaled datasets.
> > - For the plausible cases according to the first manner, LLM-as-Judge is leveraged to deeply analyze them by calling API.
> >
> > Across both lexical and semantic evaluations, LLM-generated edge descriptions are largely faithful, semantically aligned, and reliable as relational augmentation. Even among the lowest-ROUGE cases, only a small minority exhibit semantic drift, validating LLM-generated relationship descriptions as a practical and effective mechanism for transferring relational knowledge to smaller structural models. We extend it as follows.
> >
> > ### 2.1 Token-Level Lexical Similarity
> >
> > We computed several surface-form similarity metrics to quantify how closely the generated edge description matches the ground-truth term description:
> > - **ROUGE-1** and **ROUGE-L**: capturing unigram (ROUGH-1) and longest common (ROUGE-L) subsequence overlap between the given two terms’ description and LLM generated edge description.
> > - **Novelty Ratio** (i.e., 100% - token overlap ratio): assessing how much content in the generated description is newly introduced versus grounded in the original definition.
> > - **TF-IDF similarity**: measuring global lexical similarity beyond direct token overlap.
> >
> > These metrics provide a broad view of lexical alignment and potential divergence. As shown in the following tables, we can observe that LLM can bring new knowledge based on the given term description, i.e., Novelty Ratio is usually larger than 50% and ROUGH is not marginal around 20% $\sim$ 30%.
> >
> > Table. Verification of Generated Edge Description on SemEval-Food Dataset
> > | Metric              |   Mean  |   Std   |
> > |---------------------|:-------:|:-------:|
> > | ROUGE-1      |    0.3965 | 0.1091
> > | ROUGE-L    |    0.2733 | 0.0865   |
> > | Novelty Ratio     |    0.6129 | 0.1352    |
> > | TF-IDF Similarity |    0.6081 | 0.1412   |
> >
> > Table. Verification of Generated Edge Description on WordNet-Verb Dataset
> > | Metric              |   Mean  |   Std   |
> > |---------------------|:-------:|:-------:|
> > | ROUGE-1     | 0.30123 | 0.08142 |
> > | ROUGE-L      | 0.20266 | 0.06236 |
> > | Novelty Ratio       | 0.71691 | 0.10000 |
> > | TF-IDF Similarity   | 0.58944 | 0.17026 |
> >
> > Table. Verification of Generated Edge Description on MeSH Dataset
> > | Metric            |   Mean  |   Std   |
> > |-------------------|:-------:|:-------:|
> > | ROUGE-1       | 0.41018 | 0.10079 |
> > | ROUGE-L       | 0.24410 | 0.06618 |
> > | Novelty Ratio     | 0.52855 | 0.14438 |
> > | TF-IDF Similarity | 0.60339 | 0.13052 |
> >
> >
> > ### 2.2 LLM-as-Judge Semantic Consistency Evaluation
> >
> > While lexical metrics capture surface similarity, they do not fully reflect semantic alignment. Therefore, we employed an LLM-as-judge to assess whether generated descriptions retain the intended meaning of the original concept.
> >
> > To control API cost, **we selected the 80 edges with the lowest ROUGE-1 scores**, representing cases most likely to contain errors or semantic drift.
> > For each sampled edge, the LLM (GPT-4o) was prompted to evaluate semantic consistency between the generated description and the ground-truth term description. The model produced **three independent votes**, labeling each instance as:
> > - **Aligned**:  semantics are preserved.
> > - **Partially Aligned**:  minor deviations or incomplete meaning.
> > - **Misaligned**:  noticeable semantic drift or incorrect interpretation.
> >
> > Each vote included a short textual justification explaining the judgment. This process provides a more fine-grained, interpretable signal of whether the generation model introduces distortions or hallucinations beyond what surface-level metrics can detect.

---

> > > ### Author Response · Authors · 2025-11-28
> > > **Rebuttal Answer from Authors (Part V)**
> > >
> > > The template of LLM-as-Judge Prompt:
> > > ```
> > > You are a taxonomy quality reviewer. Given the canonical wiki-style node descriptions, decide whether the provided edge description is factually correct, aligned with the parent/child terms, and free of hallucinated claims.
> > >
> > > Parent Term: {parent_term}
> > > Parent Definition: {parent_definition}
> > >
> > > Child Term: {child_term}
> > > Child Definition: {child_definition}
> > >
> > > Edge Description:
> > > \"\"\"{edge_description}\"\"\"
> > >
> > > Instructions:
> > > 1. Compare the edge description against the parent/child definitions. Note any hallucinated entities or attributes that contradict the references.
> > > 2. Flag missing information that prevents you from concluding the relation.
> > > 3. Respond with a short JSON object: {{"verdict": "<Aligned|Partial|Misaligned>", "issues": ["string"], "missing_information": ["string"]}}.
> > > 4. References to placeholder pseudo nodes (e.g., "pseudo root" or "pseudo leaf") are acceptable and should not be flagged or treated as missing information solely for being placeholders or lacking extra detail.
> > > ```
> > >
> > > From the below tables, we can observe that for the low ROUGE scored edge description is not always useless. With LLM (GPT-4o) as the judge, only a small portion (around 10%) of the low ROUGE scored description is low qualified, which reflects LLM augmenting edge description is a viable manner to transfer the knowledge from LLMs to SLMs.
> > >
> > > Table. LLM-as-Judge Verification of the Lowest ROUGE 80 Generated Edge Description, on SemEval-Food Dataset
> > > | Vote          |   Ratio |
> > > |-------------------|:-------:|
> > > | Aligned |  47/80 (58.75%)|
> > > | Partial Aligned | 25/80 (31.25%)|
> > > | Misaligned | 8/80 (10%)|
> > >
> > > Table. LLM-as-Judge Verification of the Lowest ROUGE 80 Generated Edge Description, on WordNet-Verb Dataset
> > > | Vote          |   Ratio |
> > > |-------------------|:-------:|
> > > | Aligned |  36/80 (45.0%)|
> > > | Partial Aligned | 40/80 (50.0%)|
> > > | Misaligned | 4/80 (5.0%)|
> > >
> > > Table. LLM-as-Judge Verification of the Lowest ROUGE 80 Generated Edge Description, on MeSH Dataset
> > > | Vote          |   Ratio |
> > > |-------------------|:-------:|
> > > | Aligned |  58/80 (72.5%)|
> > > | Partial Aligned | 19/80 (23.75%)|
> > > | Misaligned | 3/80 (3.75%)|
> > >
> > > ---
> > >
> > > > Q2: Can the negative sampling strategy in self-supervised optimization be further detailed or ablated? Is there evidence that specific hard negative choices strongly affect generalization versus random negatives?
> > >
> > > Since this question is from W2, we answered it above.

---

### Official Review · Reviewer_Ghyo · 2025-11-01

**Soundness:** 3
**Presentation:** 2
**Contribution:** 3
**Rating:** 6
**Confidence:** 2

**Summary:**

The paper explores how SLMs can effectively perform scientific taxonomy expansion by leveraging geometric constraints and guidance from LLMs.
Authors propose SS-MONO (Structure-Semantic Monotonization), which integrates three components: (1) local taxonomy augmentation by frozen LLMs, (2) self-supervised fine-tuning of an SLM (e.g., DistilBERT) with hyperbolic constraints to preserve hierarchical transitivity, and (3) LLM-based calibration.
Experiments on SemEval-Food, MeSH, and WordNet-Verb show that SS-MONO outperforms both traditional graph-based methods (e.g., TaxoExpan, TMN, QEN) and frozen LLMs like GPT-4o-mini, establishing SLMs as cost-efficient and competitive models for scientific taxonomy expansion.

**Strengths:**

- The paper presents a novel perspective that connects hyperbolic geometry in LLM embeddings with taxonomy reasoning.
- The paper is well-motivated and technically detailed.
- The self-supervised training approach removes the need for human annotation, and the idea of “borrowing knowledge” from LLMs while retaining the efficiency of SLMs is practically appealing.
- The empirical results are comprehensive and support the claim of the paper.

**Weaknesses:**

- The impact of geometric regularization compared to simpler fine-tuning is not explained in the main text. I think it’s worth discussing the main takeaway of the ablation studies in 4.2.

**Questions:**

See weakness.

---

> ### Author Response · Authors · 2025-11-28
> **Rebuttal Answer from Authors (Part I)**
>
> Thanks very much for your review! We are excited to learn your appreciation of our hyperbolic design for effectively borrowing knowledge from LLMs to SLMs, self-supervised learning manner, and extensive experiments.
>
> Your suggestion is very actionable. We prepared the answer as follows, and we updated the paper accordingly:
> - We added Section 4.4 to systematically analyze the impact of geometric conditions with new experiments from 6 aspects.
>
> ---
>
> > W1. The impact of geometric regularization compared to simpler fine-tuning is not explained in the main text. I think it’s worth discussing the main takeaway of the ablation studies in 4.2.
>
> We agree that the impact of geometric regularization deserves clearer emphasis in the main text.
>
> In Appendix I of original submission, we prepared a few ablation studies and analysis for investigating the geometric conditions.
>
> During the rebuttal, we conducted additional experiments and upgraded the entire experiment for this part systematically.
>
> **First**, we added Section 4.4 in the main body, which detailedly discuss:
>   - the weight of $\mathcal{L}\_{structure}$ in the structure-dominated encoder as expressed in Eq. 4 for preserving the monotonicity in the hyperbolic space,
>   - and the relationship of structure loss and context loss in $\mathcal{L}\_{total}$ expressed in Eq.14.
>
> **Second**, we briefly discuss the findings of geometric conditions from other aspects in the main body and leave the clear indicator for their positions in Appendix I. You can check the new updated pdf file, and we also prepare the summarized version below:
> - The analysis for investigating the role of sequential self-attention mechanism with graph neural networks message-passing mechanism in the context-dominated encoder is in **Appendix I.2**,
>   - which suggests that the sequence-based self-attention mechanism is typically more effective than various subgraph-based message passing mechanisms;
> - The analysis of varying the number of sampled hops can be found in **Appendix I.3**,
>   - which suggests that 3-hop is a fair sampling scope and larger hop may induce noise;
> - The analysis of the difference between Euclidean and Hyperbolic manners for the structure-dominated encoder is in **Appendix I.4**,
>   - which suggests the Hyperbolic constraints is more effective to perverse the hierarchical information and help the corresponding downstream task performance;
> - The analysis of the relationship between the hard negative sampling and random sampling is in **Appendix I.5**,
>   - which suggest the negative hard sampling can bring additional performance gain but large volume of negative hard samples can ruin the performance.

---

### Official Review · Reviewer_orUz · 2025-11-01

**Soundness:** 2
**Presentation:** 3
**Contribution:** 2
**Rating:** 4
**Confidence:** 3

**Summary:**

The paper introduces SS-MONO (Structure-Semantic Monotonization), a novel pipeline for efficient and accurate scientific taxonomy expansion. Recognizing the strong hyperbolicity in Large Language Model (LLM) embeddings, SS-MONO addresses the high computational cost and hallucination issues associated with directly using LLMs on domain-specific taxonomies. SS-MONO leverages LLM augmentation and distills this knowledge into Small Language Models (SLMs) through self-supervised fine-tuning enforced by geometric constraints. Key modules include a structure-dominated encoder using hyperbolic representation learning to preserve hierarchy (monotonicity) and a context-dominated encoder for contextual semantics. Experiments show that a fine-tuned SLM (e.g., DistilBERT-base-110M) consistently outperforms frozen LLMs and deep learning baselines on expansion tasks.

**Strengths:**

1. Efficiency and Cost-Effectiveness

SS-MONO implements an LLM-to-SLM distillation approach, fine-tuning a Small Language Model (SLM), such as DistilBERT-base-110M. This strategy provides a practical and economical alternative to the high computational costs and difficulties associated with directly using Large Language Models (LLMs) on domain-specific taxonomies.

2. Structural Integrity and Superior Overall Performance

The pipeline utilizes hyperbolic representation learning in its structure-dominated encoder to enforce geometric constraints (monotonicity), which preserves the hierarchical order. This structural awareness allows the fine-tuned SLM to consistently outperform frozen LLMs and domain-specific deep learning baselines overall.

3. Self-Supervised Training

The entire training process is self-supervised, guided by the existing taxonomy's topology. This design eliminates the need for expensive human labeling efforts for expansion tasks

**Weaknesses:**

1. Performance on Large-Scale Taxonomies

The methodology demonstrates significantly lower effectiveness on the large-scale WordNet-Verb dataset (13,936 nodes, depth 12) compared to smaller datasets. SS-MONO's overall average ranking on WordNet-Verb is 1626.52, which is not SOTA and substantially less favorable than its performance on SemEval-Food (239.17) and MeSH (436.82). Does this indicate that the proposed method could have issues in scalability and can be less effective on large-scale Taxonomies?

2. Non-Leaf Volatility with LLM Augmentation Observed

The performance metrics show that including LLM Augmented Descriptions (AD) does not always enhance intermediate (non-leaf) expansion. For example, on WordNet-Verb, the Non-leaf R@10 metric decreases sharply from 0.099 (SS-MONO w/o AD) to 0.035 (full SS-MONO). Other datasets have the same observation. What could lead to the performance difference between leaf nodes and non-leaf nodes? Is it because non-leaf nodes have more complicated relationship and LLM can hardly have clear annotations?

**Questions:**

1. Please update the bold format in all the experiment tables in the paper carefully. Currently many bolded numbers in the table are not exactly the one with the best value. This is very confusing and could lead to wrong conclusion.

2. Please address the above two concerns I have.

---

> ### Author Response · Authors · 2025-11-28
> **Rebuttal Answer from Authors (Part I)**
>
> Dear Reviewer orUz,
>
> Thanks very much for your review! We are excited to learn of your acknowledgement of our LLM-to-SLM distillation idea, geometric-constraint design, self-supervised learning manner, and promising empirical performance.
>
> For each of your raised concerns, we carefully prepared the answer as follows.
>
> Also, we updated the paper with new content under the yellow background color:
> - We updated Section 4.2 to deeply discuss how to analyze the performance of an individual algorithm across different datasets
> - We added Appendix O to discuss the current limitation of generated edge description with the structural condition in terms of leaf and non-leaf positions, and list a few interesting future research directions.
>
> ---
>
> > **W1**. Performance on Large-Scale Taxonomies. The methodology demonstrates significantly lower effectiveness on the large-scale WordNet-Verb dataset (13,936 nodes, depth 12) compared to smaller datasets. SS-MONO's overall average ranking on WordNet-Verb is 1626.52, which is not SOTA and substantially less favorable than its performance on SemEval-Food (239.17) and MeSH (436.82). Does this indicate that the proposed method could have issues in scalability and can be less effective on large-scale Taxonomies?
>
> We appreciate the reviewer’s concern regarding the higher Mean Rank (MR) on the large-scale WordNet-Verb dataset. Please allow me to clarify that the proposed method does have issues in scalability or is less effective on large-scale taxonomies. The core reason is that MR is an absolute but not normalized value.
>
> We extend the detailed reasoning below, and **they are added into Section 4.2**.
>
> We agree that WordNet-Verb (13,936 nodes, depth 12) is substantially more challenging than SemEval-Food and MeSH because:
> - the candidate space is an order of magnitude larger,
> - the hierarchy is deeper,
> - and many verbs are highly polysemous and sparsely linked.
>
> These factors naturally inflate absolute MR values, so **MR values across datasets are not directly comparable**. Because MR is not normalized by the number of candidate positions, its absolute value grows naturally with larger and deeper taxonomies and then cannot be directly compared across datasets with divergent scales.
>
> For cross-dataset comparisons of an individual algorithm (including ours and baselines), **MRR (Mean ​​Reciprocal Rank)** and **Recall@k are more appropriate indicators**, as they are scale-invariant and reflect relative ranking quality independent of taxonomy size. Based on that, SS-Mono remains better than prior methods in terms of relative improvement, in Table 2, SS-Mono usually achieves the first place in MRR and Recall@1 across three datasets.
>
> In summary, above analysis suggests that SS-Mono does not fundamentally break down on large taxonomies; instead, it faces the same inherent difficulty as the much larger, deeper search space. **Since MRR and Recall@k are metrics normalized by candidate set size**.
>
> ---
>
> > **W2**. Non-Leaf Volatility with LLM Augmentation Observed. The performance metrics show that including LLM Augmented Descriptions (AD) does not always enhance intermediate (non-leaf) expansion. For example, on WordNet-Verb, the Non-leaf R@10 metric decreases sharply from 0.099 (SS-MONO w/o AD) to 0.035 (full SS-MONO). Other datasets have the same observation. What could lead to the performance difference between leaf nodes and non-leaf nodes? Is it because non-leaf nodes have more complicated relationship and LLM can hardly have clear annotations?
>
> We appreciate the reviewer’s observation regarding the differing effects of LLM-Augmented Descriptions (AD) on leaf vs. non-leaf expansion.
>
> Next, we explain the latent reasoning for this difference, **which is added into Appendix O**.
>
> Our analysis suggests that the phenomenon stems from how LLM Augmented Descriptions (AD) interact with the structural roles of different node types in a taxonomy from three aspects.
>
> 1. **Leaf-node descriptions tend to be structurally consistent and aligned with the insertion task**.
>
> For leaf nodes, the LLM-generated description typically focuses on the concept’s _global semantic role_, for example, that it represents a terminal category or a fine-grained subtype. Because leaf nodes have no children, the LLM’s descriptions are naturally concise and homogeneous across examples. This consistency gives the scorer a clearer signal about where the node belongs in the taxonomy.

---

> > ### Author Response · Authors · 2025-11-28
> > **Rebuttal Answer from Authors (Part II)**
> >
> > 2. **Non-leaf descriptions exhibit higher diversity due to dual relationships**.
> >
> > In contrast, non-leaf nodes participate in both upward (to parents) and downward (to children) relationships. Their descriptions therefore need to capture more heterogeneous information. Empirically, we observe that LLMs produce a broader variety of relational phrasing for non-leaf nodes, since the prompts must describe _how two or more concepts relate_ rather than how a single concept fits into the taxonomy. This diversity increases variability in the augmented text representations.
> >
> > 3. **This variability interacts with fine-grained structural ranking and can dilute local signals**.
> >
> > Non-leaf insertion is a more fine-grained ranking task because intermediate positions are surrounded by more structurally similar alternatives. When the textual descriptions exhibit higher variance, the scorer may receive weaker or less discriminative signals about subtle differences within the local subgraph. In contrast, leaf nodes benefit from the more uniform and taxonomy-aligned global descriptions.
> >
> > To sum up, LLM Augmented Descriptions is so far a viable solution which provides the additional knowledge for the existing taxonomy and can be leveraged to improve the SLM’s performance based on our designed geometric deep learning constraints and self-supervised learning manner, as the effectiveness shown in the extensive experiments.
> >
> > Indeed, it is not perfect and has the latent drawback as discussed above, and we are more than willing to set it as our future research direction, a few possible directions are listed below:
> > - **Incorporating external web knowledge for richer edge descriptions.**
> > Future work can enable controlled web-search integration so the LLM can retrieve concrete, verifiable facts about concept relationships. This can produce more accurate and informative edge-oriented descriptions, especially for non-leaf nodes whose semantics depend on multiple relational contexts.
> > - **Leveraging local neighborhood structure during AD generation.**
> > Integrating information from the node’s local subgraph—such as siblings, parents, children, and subtree summaries—can anchor the generated descriptions to the actual taxonomy structure. This helps reduce semantic drift and lowers variance in non-leaf descriptions.
> > - **Developing multi-step, grounded generation pipelines.**
> > A more robust AD pipeline can combine (1) local-structure grounding, (2) retrieved external knowledge, and (3) a self-critique/refinement step. Such multi-step generation aims to stabilize the textual signal, filter inconsistent relational statements, and produce higher-fidelity and richer descriptions that better support fine-grained non-leaf insertion.
> >
> >
> > ---
> >
> >
> > > **Q1**. Please update the bold format in all the experiment tables in the paper carefully. Currently many bolded numbers in the table are not exactly the one with the best value. This is very confusing and could lead to wrong conclusion.
> >
> > Thanks for the consideration.
> >
> > First, we have used different colors to highlight the performance.
> >
> > Second, we have rewritten the entire Section 4 for better readability, and use yellow background color to mark the updated part for your reference.
> >
> > ---
> >
> > > **Q2**. Please address the above two concerns I have.
> >
> > Thanks. We have addressed them above and we are more than excited to hear your feedback.

---

### Official Review · Reviewer_7uga · 2025-11-06

**Soundness:** 2
**Presentation:** 2
**Contribution:** 2
**Rating:** 6
**Confidence:** 3

**Summary:**

This paper addresses the task of taxonomy expansion in scientific domains, specifically adding new concept nodes to an existing taxonomy (precisely in a directed acyclic hierarchy of concepts). The paper proposes a pipeline that borrows knowledge from an LLM and transfers it into SLM, which can then perform the taxonomy expansion efficiently. SS-MONO has three stages: (1) local taxonomy augmentation using an LLM - for each candidate insertion position in the taxonomy (parent node and a child node between which a new concept might be inserted) a pretained LLM is prompted to generate a textual explanation or description of local context (semantic features about the candidate position), (2) fine-tuning a SLM with geometric constraints to rank candidate insertion positions for a query concept (structure-dominated encoder that projects concept embeddings into hyperbolic space by nested entailment cones to keep hierarchical relationships, and a context-dominated encoder that embeds the textual descriptions) - here a key idea to enforce the monotonic ordering (child ≼ query ≼ parent in the embedding space) and (3) LLM-based calibrationof SLM score where to insert the new concept with use of second LLM call to re-rank the top-$k$ predicted positions. Experiments on three benchmarks demonstrate, in some cases, superiority and advantages of SS-MONO over other compared methods.

**Strengths:**

- The paper presents a creative integration of ideas from different domains: it combines hyperbolic geometry (for representing hierarchical structure) with LLM-based semantic augmentation in a small-model pipeline. While hyperbolic embeddings have been used for hierarchical tasks before and others have leveraged language models for taxonomy expansion, this provides a limited but still novel work in how it brings these together. Especially the notion of “Structure-Semantic Monotonization” is a novel formulation to ensure monotonicity in latent space (with use of nested entailment cones to enforce transitivity).
- The technical quality of the work appears high. The paper is thorough in justifying and evaluating the approach.
- The training scheme is cleverly self-supervised, avoiding manual annotations (removal of nodes predict their insertion).

**Weaknesses:**

- Ironically, a method motivated by avoiding LLM usage still depends on LLMs at key points. The small model alone does a lot of the work, but the pipeline requires a capable LLM to provide the augmented descriptions and to perform final calibration. In the ablation without LLM augmentations (SS-MONO w/o AD), the small model’s performance, although competitive, is not clearly superior to the best prior methods.
- The paper does not delve deeply into what the LLM-generated “textual explanations” look like or how consistent their quality is. This is a bit of a black box in the description. If the LLM outputs poor or hallucinated explanations for some candidate positions, does that ever confuse the SLM during training? One could imagine the LLM sometimes generating a misleading context (especially if the taxonomy contains very specialized terms that the LLM isn’t familiar with). The authors did not mention any filtering or human verification of the LLM outputs. It would strengthen the work to either demonstrate that these augmented descriptions are almost always accurate, or to describe measures to handle noise in those descriptions.
- Due to my understanding, there is no comparison to one of the baseline scenarios when LLM are similarly asked in turns to perform similar steps lika SS-MONO (pair insertion evaluairion, re-ranking).

**Questions:**

- Can the authors provide more details or examples of the prompt and output used for the LLM when generating the candidate position descriptions? This is currently abstract in the paper. For instance, if the candidate position is between parent concept P and child concept C, do you prompt the LLM with something like “Explain the relationship between P and its subcategory C” or “Give a description of where C fits under P”? And does the LLM output a few sentences describing that taxonomic context? An example would help in understanding what knowledge the LLM is injecting. Additionally, did you observe any instances of the LLM outputting incorrect information about the taxonomy? If so, how did you mitigate that (e.g., do you simply trust whatever the LLM says, or do you have a way to sanity-check it)? Clarifying this will help assess the reliability of the augmentation. And could you please evaluate LLM with the same procedure (subtasks) as SLM (inference).
- You introduce a complex hyperbolic constraint system for the structure-dominated encoder. Did you compare or ablate this against a simpler approach (for example, a Euclidean encoder or a transformer-based graph encoder without hyperbolic projection)? In Appendix I.1 you mention investigating the role of “geometric deep learning” – can you summarize those findings? It would be insightful to know how much the hyperbolic embedding improved things. Perhaps the model could also be trained in Euclidean space with a learned ordering constraint – would that fail or perform worse?
- In cases where a concept has multiple true parent locations (non-leaf multi-attachments), how would you operationally use SS-MONO to attach it in all the correct places?

---

> ### Author Response · Authors · 2025-11-28
> **Rebuttal Answer from Authors (Part I)**
>
> Dear Reviewer 7uga,
>
> Thanks very much for your review! We are excited to know you like our hyperbolic geometry idea on small LLMs and our design of “Structure-Semantic Monotonization”.
>
> We take your review very seriously and prepared the answer for each concern below.
>
> Moreover, we have updated the paper according to your comments, marked within yellow background color. In brief, we added:
> - Appendix M to systematically analyze LLM-augmented edge description with various check manners.
> - Appendix N to analyze the performance of directly fine-tuning LLMs with our LLM-to-SLM distillation design.
> - Appendix I.4 to compare the Euclidean encoding with the hyperbolic encoding.
>
> Next, we answer each raised concern deeply.
>
> ---
>
> > **W1**. Ironically, a method motivated by avoiding LLM usage still depends on LLMs at key points. The small model alone does a lot of the work, but the pipeline requires a capable LLM to provide the augmented descriptions and to perform final calibration. In the ablation without LLM augmentations (SS-MONO w/o AD), the small model’s performance, although competitive, is not clearly superior to the best prior methods.
>
> **First**, please allow us to clarify that **we are not attempting to avoid LLMs at all, but investigating how to wisely use LLMs to guide SLMs in an affordable way**. To be more specific:
>
> As stated from lines 98 to 107, we are working on:
> - how to `borrow knowledge’ from LLMs to SLMs
> - the `borrowing’ process should avoid computational cost as much as possible
>
> Therefore, we propose to lightly use LLM to augment a small part of taxonomy just through LLM’s inference, and design the hyperbolic-aware Structure-Semantic Monotonization objective to fine-tune SLM to reach the competitive capabilities. The philosophy of using LLMs to guide SLMs is common and widely-adopted, avoiding LLMs at all seems ideal but not a quite ready nor feasible direction so far [1, 2].
>
> **Second**, your observation is correct that _``In the ablation without LLM augmentations (SS-MONO w/o AD), the small model’s performance, although competitive, is not clearly superior to the best prior methods.’’_
>
> This observation just demonstrates two of our main points:
> - Our hyperbolic design is effective, i.e., even without LLMs augmentation edge explanation, it can still achieve competitive but not best performance
> - LLM’s lightweight usage is indispensable for helping SLMs achieve the best performance
>
> Reference:
>
> [1] He et al, Harnessing Explanations: LLM-to-LM Interpreter for Enhanced Text-Attributed Graph Representation Learning. ICLR 2024
>
> [2] Belcak et al., Small Language Models are the Future of Agentic AI. arXiv 2025

---

> ### Author Response · Authors · 2025-11-28
> **Rebuttal Answer from Authors (Part II)**
>
> > **W2** (and **Q1**). The paper does not delve deeply into what the LLM-generated “textual explanations” look like or how consistent their quality is. This is a bit of a black box in the description. If the LLM outputs poor or hallucinated explanations for some candidate positions, does that ever confuse the SLM during training? One could imagine the LLM sometimes generating a misleading context (especially if the taxonomy contains very specialized terms that the LLM isn’t familiar with). The authors did not mention any filtering or human verification of the LLM outputs. It would strengthen the work to either demonstrate that these augmented descriptions are almost always accurate, or to describe measures to handle noise in those descriptions.
>
> **First**, we provide examples of our prompt template along with sample inputs and outputs. **Second**, we present the results of various sanity checks, which demonstrate that the LLM generated edge description is trustworthy. **The below answer is now included in Appendix M**.
>
> ## 1. Edge Description Generation Process and Example
> The edge description is designed to explicitly describe the relationship between a hypernym and its hyponym. This additional relational context enriches the information available to the model beyond term descriptions alone. To minimize the risk of hallucination or unsupported claims, we incorporated explicit constraints within the prompt instructions. The full prompt template for generating LLM-based edge descriptions is shown below.
> ```
> Known parents of {parent_name}: {parent_context}
> Child term: {child_name}
> Child definition: {child_definition},
> Known children of {child_name}: {child_context}",
>
> Task: Write a concise relationship explanation (2-4 sentences) describing how the child relates to the parent. Ground every statement in the provided definitions/context and avoid inventing unsupported facts. Mention at least one concrete trait that ties the child specifically to the parent category.
> ```
>
> ### 1.1 Example edge description generated by LLMs:
>
> Input:
> - Parent node, i.e., 445: dairy product: dairy product is milk and butter and cheese
> - Child node, i.e.,  474: double creme: double creme is cream with a fat content of 48% or more
>
> Output:
> - Edge Description: {"445-474": "Double creme is a type of dairy product because it is made from cream, which falls under the broader category of dairy products as defined by its inclusion in \"milk and butter and cheese.\"  \n\nSpecifically, double creme's high fat content (48% or more) distinguishes it within the range of dairy products."}
>
> ## 2. Sanity Check
>
> To further assess the quality and faithfulness of the generated edge descriptions, we conducted **a systematic sanity check** comparing each generated description with its corresponding term description in the original taxonomy dataset.
>
> This validation step consists of two complementary analyses: **token-level lexical similarity** and **LLM-based semantic consistency evaluation (LLM-as-Judge)**.
> - The first one is for measuring from the token level for the capability of LLM to maintain old knowledge and bring new knowledge, which is efficient for scaled datasets.
> - For the plausible cases according to the first manner, LLM-as-Judge is leveraged to deeply analyze them by calling API.
>
> Across both lexical and semantic evaluations, LLM-generated edge descriptions are largely faithful, semantically aligned, and reliable as relational augmentation. Even among the lowest-ROUGE cases, only a small minority exhibit semantic drift, validating LLM-generated relationship descriptions as a practical and effective mechanism for transferring relational knowledge to smaller structural models. We extend as follows.
>
> ### 2.1 Token-Level Lexical Similarity
>
> We computed several surface-form similarity metrics to quantify how closely the generated edge description matches the ground-truth term description:
> - **ROUGE-1** and **ROUGE-L**: capturing unigram (ROUGH-1) and longest common (ROUGE-L) subsequence overlap between the given two terms’ description and LLM generated edge description.
> - **Novelty Ratio** (i.e., 100% - token overlap ratio): assessing how much content in the generated description is newly introduced versus grounded in the original definition.
> - **TF-IDF similarity**: measuring global lexical similarity beyond direct token overlap.
>
> These metrics provide a broad view of lexical alignment and potential divergence. As shown in the following tables, we can observe that LLM can bring new knowledge based on the given term description, i.e., Novelty Ratio is usually larger than 50% and ROUGH is not marginal around 20% $\sim$ 30%.
>
> Table. Verification of Generated Edge Description on SemEval-Food Dataset
> | Metric              |   Mean  |   Std   |
> |--------------|:-------:|:-------:|
> | ROUGE-1      |    0.3965 | 0.1091
> | ROUGE-L    |    0.2733 | 0.0865   |
> | Novelty Ratio     |    0.6129 | 0.1352    |
> | TF-IDF Similarity |    0.6081 | 0.1412   |

---

> ### Author Response · Authors · 2025-11-28
> **Rebuttal Answer from Authors (Part III)**
>
> Table. Verification of Generated Edge Description on WordNet-Verb Dataset
> | Metric              |   Mean  |   Std   |
> |---------------------|:-------:|:-------:|
> | ROUGE-1     | 0.30123 | 0.08142 |
> | ROUGE-L      | 0.20266 | 0.06236 |
> | Novelty Ratio       | 0.71691 | 0.10000 |
> | TF-IDF Similarity   | 0.58944 | 0.17026 |
>
> Table. Verification of Generated Edge Description on MeSH Dataset
> | Metric            |   Mean  |   Std   |
> |-------------------|:-------:|:-------:|
> | ROUGE-1       | 0.41018 | 0.10079 |
> | ROUGE-L       | 0.24410 | 0.06618 |
> | Novelty Ratio     | 0.52855 | 0.14438 |
> | TF-IDF Similarity | 0.60339 | 0.13052 |
>
>
> ### 2.2 LLM-as-Judge Semantic Consistency Evaluation
> While lexical metrics capture surface similarity, they do not fully reflect semantic alignment. Therefore, we employed an LLM-as-judge to assess whether generated descriptions retain the intended meaning of the original concept.
> To control API cost, **we selected the 80 edges with the lowest ROUGE-1 scores**, representing cases most likely to contain errors or semantic drift.
> For each sampled edge, the LLM (GPT-4o) was prompted to evaluate semantic consistency between the generated description and the ground-truth term description. The model produced **three independent votes**, labeling each instance as:
> - **Aligned**:  semantics are preserved.
> - **Partially Aligned**:  minor deviations or incomplete meaning.
> - **Misaligned**:  noticeable semantic drift or incorrect interpretation.
>
> Each vote included a short textual justification explaining the judgment. This process provides a more fine-grained, interpretable signal of whether the generation model introduces distortions or hallucinations beyond what surface-level metrics can detect.
>
> The template of LLM-as-Judge Prompt:
> ```
> You are a taxonomy quality reviewer. Given the canonical wiki-style node descriptions, decide whether the provided edge description is factually correct, aligned with the parent/child terms, and free of hallucinated claims.
>
> Parent Term: {parent_term}
> Parent Definition: {parent_definition}
>
> Child Term: {child_term}
> Child Definition: {child_definition}
>
> Edge Description:
> \"\"\"{edge_description}\"\"\"
>
> Instructions:
> 1. Compare the edge description against the parent/child definitions. Note any hallucinated entities or attributes that contradict the references.
> 2. Flag missing information that prevents you from concluding the relation.
> 3. Respond with a short JSON object: {{"verdict": "<Aligned|Partial|Misaligned>", "issues": ["string"], "missing_information": ["string"]}}.
> 4. References to placeholder pseudo nodes (e.g., "pseudo root" or "pseudo leaf") are acceptable and should not be flagged or treated as missing information solely for being placeholders or lacking extra detail.
> ```
> From the below tables, we can observe that for the low ROUGE scored edge description is not always useless. With LLM (GPT-4o) as the judge, only a small portion (around 10%) of the low ROUGE scored description is low qualified, which reflects LLM augmenting edge description is a viable manner to transfer the knowledge from LLMs to SLMs.
>
> Table. LLM-as-Judge Verification of the Lowest ROUGE 80 Generated Edge Description, on SemEval-Food Dataset
> | Vote          |   Ratio |
> |-------------------|:-------:|
> | Aligned |  47/80 (58.75%)|
> | Partial Aligned | 25/80 (31.25%)|
> | Misaligned | 8/80 (10%)|
>
> Table. LLM-as-Judge Verification of the Lowest ROUGE 80 Generated Edge Description, on WordNet-Verb Dataset
> | Vote          |   Ratio |
> |-------------------|:-------:|
> | Aligned |  36/80 (45.0%)|
> | Partial Aligned | 40/80 (50.0%)|
> | Misaligned | 4/80 (5.0%)|
>
> Table. LLM-as-Judge Verification of the Lowest ROUGE 80 Generated Edge Description, on MeSH Dataset
> | Vote          |   Ratio |
> |-------------------|:-------:|
> | Aligned |  58/80 (72.5%)|
> | Partial Aligned | 19/80 (23.75%)|
> | Misaligned | 3/80 (3.75%)|
>
> ---
>
> > **W3**. Due to my understanding, there is no comparison to one of the baseline scenarios when LLM are similarly asked in turns to perform similar steps like SS-MONO (pair insertion evaluation, re-ranking).
>
> The suggested comparison, i.e., prompting LLMs to perform pairwise insertion evaluation followed by re-ranking, was considered in the original submission. However, in practice, LLMs struggle when asked to rank over the full candidate space: their recall@k and precision@k metrics are nearly zero, as shown in Table 2. Because the ground-truth parent positions almost never appear in the top-k list, a re-ranking step cannot provide meaningful calibration or improvement. As a result, such a baseline would not offer a fair or informative comparison.

---

> ### Author Response · Authors · 2025-11-28
> **Rebuttal Answer from Authors (Part IV)**
>
> To deeply address your concern, we incorporate an LLM as the pretrained encoder and fine-tune it inside our pipeline. We report these results to demonstrate the role LLM representations can play when integrated more directly into our framework. **The below content is added into the paper Appendix N**.
>
> To test the adaptability of our pipeline, we adopt TinyLlama-1.1B-intermediate-step-1431k-3T [1] as a representative lightweight LLM and attach a LoRA adapter [2] to the last four transformer layers due to computational resource constraints. Specifically, we fine-tune the attention and feed-forward projection modules, including q_proj, k_proj, v_proj, o_proj, gate_proj, up_proj, and down_proj, while freezing the token embedding layer. TinyLlama-1.1B is fine-tuned under the same learning objective as the original SS-Mono, employing the loss function in Equation 14 to ensure a consistent optimization setup for comparison.
>
> This experiment further confirms that our geometric deep learning objective is easily compatible with off-the-shelf LLM checkpoints: only minimal modifications are needed to plug in an LLM encoder, and training remains computationally lightweight.
>
> In the below table, the results show comparable mean ranks (MR and MRR) across all testing nodes and better results for leaf node insertions; however, worse results for non-leaf node insertions. The empirical results show that LLM fine-tuning does not necessarily lead to consistent improvements.
>
> As shown below, the fine-tuned TinyLlama variant achieves comparable MR and MRR and notably improves leaf insertions, but it degrades performance on non-leaf nodes. These results suggest that our LLM-SLM-distillation design is not worse than (or even better than) LLM fine-tuning.
>
> Table. Performance of SS-Mono with Fine-Tuned LLM on SemEval-Food Dataset
> | Method            |  Total  |        |        |        |         |        |        |         |    Leaf   |         |         |  Non-leaf |         |         |
> |:-----------------:|:-------:|:------:|:------:|:------:|:-------:|:------:|:------:|:-------:|:---------:|:-------:|:-------:|:---------:|:-------:|:-------:|
> |                   |  MR(↓)  | MRR(↑) | R@1(↑) | R@5(↑) | R@10(↑) | P@1(↑) | P@5(↑) | P@10(↑) |   MR(↓)   |  MRR(↑) | R@10(↑) |   MR(↓)   |  MRR(↑) | R@10(↑) |
> | SS-Mono | **239.169** | **0.4**    | **0.186**  | **0.299**  | **0.325**   | **0.392**  | **0.126**  | **0.068**   | 143.93667 | 0.70478 | 0.64539 | **756.73471** | **0.14665** | **0.05882** |
> | Fine-Tuned TinyLlama    | 253.911 | 0.373  | 0.122  | 0.241  | 0.309   | 0.257  | 0.101  | 0.065   | **73.851**    | **0.754**   | **0.652**   | 1139.808  | 0.08    | 0.045   |
>
> Reference:
>
> [1] Zhang et al., Tinyllama: An open-source small language model, arXiv 2024
>
> [2] Hu et al., LoRA: Low-Rank Adaptation of Large Language Models, ICLR 2022
>
> ---
>
> > **Q1** is from **W2**
>
> We answered Q1 in W2 above.
>
> ---
>
> > **Q2**. You introduce a complex hyperbolic constraint system for the structure-dominated encoder. Did you compare or ablate this against a simpler approach (for example, a Euclidean encoder or a transformer-based graph encoder without hyperbolic projection)? In Appendix I.1 you mention investigating the role of “geometric deep learning” – can you summarize those findings? It would be insightful to know how much the hyperbolic embedding improved things. Perhaps the model could also be trained in Euclidean space with a learned ordering constraint – would that fail or perform worse?
>
> Thanks for your consideration. In the answer below, (1) we design the Euclidean encoder without hyperbolic projection that is **added in Appendix I.4**, (2) and further clarify the ablation study in Appendix I.1.
>
> ## 1. Euclidean transformer-based encoder
> To more thoroughly assess whether the hyperbolic encoder and cone loss are essential as suggested, we further conduct an ablation in which the hyperbolic encoder is replaced by a transformer-based encoder, and the original cone loss is replaced by a Euclidean ordering-constraint objective as follows:
> $L = \tfrac{1}{2}\big[\max(0, d(q,p) - d(p,c) + m) + \max(0, d(q,c) - d(p,c) + m)\big]$
>
> In the above Euclidean encoder, the distance function $d(\cdot)$ leverages the L2 norm. Based on this, for the structural encoder, we retain the same 2 transformer layers used in SS-Mono but eliminate the hyperbolic components, including both the Euclidean-to-hyperbolic projection and the cone scoring mechanism. After that, the representations from the structural encoder are used directly as inputs to the Euclidean ordering-constraint objective, as shown above, ensuring consistent comparisons while isolating the hyperbolic structural encoder and cone score.
>
> As shown in the results below, SS-Mono performs better across all metrics in the Total, Leaf, and Non-leaf settings across three datasets, demonstrating the effectiveness of the hyperbolic structural encoder and the cone-based objective.

---

> > ### Author Response · Authors · 2025-11-28
> > **Rebuttal Answer from Authors (Part V)**
> >
> > Table. Performance of SS-Mono with Hyperbolic Encoder and Euclidean Encoder on SemEval-Food Dataset
> > | Method            |   Total  |        |        |        |         |        |        |         |    Leaf   |         |         |  Non-leaf  |         |         |
> > |:-----------------:|:--------:|:------:|:------:|:------:|:-------:|:------:|:------:|:-------:|:---------:|:-------:|:-------:|:----------:|:-------:|:-------:|
> > |                   |   MR(↓)  | MRR(↑) | R@1(↑) | R@5(↑) | R@10(↑) | P@1(↑) | P@5(↑) | P@10(↑) |   MR(↓)   |  MRR(↑) | R@10(↑) |    MR(↓)   |  MRR(↑) | R@10(↑) |
> > | SS-Mono (Hyperbolic) | **239.169**  | **0.400**    | **0.186**  | **0.299**  | **0.325**   | **0.392**  | **0.126**  | **0.068**   | **143.937** | **0.705** | **0.645** | **756.735**  | **0.147** | 0.059 |
> > | SS-Mono (Euclidean)    | 654.712  | 0.175  | 0.019  | 0.077  | 0.100     | 0.041  | 0.032  | 0.021   | 263.317   | 0.298   | 0.148   | 2580.372   | 0.080    | **0.062**   |
> >
> >
> > Table. Performance of SS-Mono with Hyperbolic Encoder and Euclidean Encoder on WordNet-Verb Dataset
> > | Method            |   Total  |        |        |        |         |        |        |         |    Leaf   |         |         |  Non-leaf  |         |         |
> > |:-----------------:|:--------:|:------:|:------:|:------:|:-------:|:------:|:------:|:-------:|:---------:|:-------:|:-------:|:----------:|:-------:|:-------:|
> > |                   |   MR(↓)  | MRR(↑) | R@1(↑) | R@5(↑) | R@10(↑) | P@1(↑) | P@5(↑) | P@10(↑) |   MR(↓)   |  MRR(↑) | R@10(↑) |    MR(↓)   |  MRR(↑) | R@10(↑) |
> > | SS-Mono (Hyperbolic)    | **1626.522** | **0.334**  | **0.106**  | **0.208**  | **0.260**    | **0.163**  | **0.064**  | **0.040**    | **922.541** | **0.521** | **0.457** | **4551.311** | **0.122** | **0.035** |
> > | SS-Mono (Euclidean)        | 3682.875 | 0.059  | 0.005  | 0.018  | 0.025   | 0.007  | 0.006  | 0.004   | 1362.167  | 0.074   | 0.021   | 13641.050   | 0.042   | 0.030    |
> >
> > Table. Performance of SS-Mono with Hyperbolic Encoder and Euclidean Encoder on MeSH Dataset
> > | Method            |   Total  |        |        |        |         |        |        |         |    Leaf   |         |         |  Non-leaf  |         |         |
> > |:-----------------:|:--------:|:------:|:------:|:------:|:-------:|:------:|:------:|:-------:|:---------:|:-------:|:-------:|:----------:|:-------:|:-------:|
> > |                   |   MR(↓)  | MRR(↑) | R@1(↑) | R@5(↑) | R@10(↑) | P@1(↑) | P@5(↑) | P@10(↑) |   MR(↓)   |  MRR(↑) | R@10(↑) |    MR(↓)   |  MRR(↑) | R@10(↑) |
> > | SS-Mono (Hyperbolic)  | **436.820**   | **0.427**  | **0.074**  | **0.197**  | **0.288**   | **0.173**  | **0.093**  | **0.068**   | **390.717** | **0.570** | **0.476** | **540.551**  | **0.334** | **0.166** |
> > | SS-Mono (Euclidean)          | 8976.253 | 0.075  | 0.010   | 0.046  | 0.061   | 0.023  | 0.021  | 0.014   | 7038.222  | 0.039   | 0.026   | 13490.450   | 0.104   | 0.089   |
> >
> > ## 2. Clarification of ablation study in Appendix I.1
> >
> > **First**, as you pointed out in Appendix I.1 (now indexed as Section 4.4), we did an ablation study analyzing the weights of cone loss. Appendix I.1 (now indexed as Section 4.4) detailedly discuss:
> >   - (1) the importance of the weight of $\mathcal{L}_{structure}$ in the structure-dominated encoder as expressed in Eq. 4 for preserving the monotonicity in the hyperbolic space.
> >     - In Table 4, it can be observed that totally removing the structure-dominated encoder of cone (i.e., weight = 0) usually induces the worst performance.
> >   - (2) and the relationship of structure loss and context loss in $\mathcal{L}_{total}$ expressed in Eq.14.
> >     - According to Eq.14, we have $\beta$ for all sampled neighbors in general, $\mu$ for sampled descendants only, $\lambda$ for sampled ancestors only, and $\xi$ for sampled siblings only.
> >     - In Table 5, we can observe that with all sampled nodes considered together, i.e., weight = 1111, the optimal results are obtained, compared with any ablation.

---

> ### Author Response · Authors · 2025-11-28
> **Rebuttal Answer from Authors (Part VI)**
>
> **Second**, during the rebuttal, we conducted additional experiments and upgraded the entire experiment for this part systematically. We briefly discuss the findings of geometric conditions from other aspects in the main body and leave the clear indicator for their positions in Appendix I.
>
> You can check the new updated pdf file, and we also prepare the summarized version below:
>   - Analysis for investigating the role of self-attention with graph neural networks in the context-dominated encoder is in **Appendix I.2**,
>     - which suggests that the sequence-based self-attention mechanism is typically more effective than various subgraph-based message passing mechanisms;
>   - Analysis of the sampling of the number of hops can be found in **Appendix I.3**,
>     - which suggests that 3-hop is a fair sampling scope and larger hop may induce noise;
>   - Analysis of the difference between Euclidean and Hyperbolic manners for the structure-dominated encoder is in **Appendix I.4**,
>     - which suggests the Hyperbolic constraints is more effective to perverse the hierarchical information and help the corresponding downstream task performance;
>   - Analysis of the relationship between the hard negative sampling and random sampling is in **Appendix I.5**,
>     - which suggest the negative hard sampling can bring additional performance gain but large volume of negative hard samples can ruin the performance;
>
> ---
>
> > Q3. In cases where a concept has multiple true parent locations (non-leaf multi-attachments), how would you operationally use SS-MONO to attach it in all the correct places?
> In general, for concept nodes that legitimately attach to multiple parent-child positions (i.e., non-leaf multi-attachments), SS-MONO evaluates each gold parent independently. To be specific,
>
> **How SS-Mono Operates**: Let $c$ denote the concept and $ \lbrace p\_1, \dots, p\_k \rbrace $ denote its gold parent set. Each ($c$, $p_i$) pair is treated as a separate ranking instance.
>
> **How to Evaluate**: Therefore, the evaluation is ranking-based: we compute (1) the rank assigned to each gold parent within the model’s predicted positions, and (2) whether each gold parent appears among the top-k predictions.
> - Under this formulation, a model may exhibit high precision while still yielding a higher mean rank (MR). This pattern arises when the model strongly recovers one of the true parent positions, boosting precision, while failing to rank the remaining true parents highly, thereby inflating the MR.
> - The `Recall@k` (or `R@k`) metric is normalized by the number of ground truth insertions, which could be another indicator of whether the model performs well across multiple ground truth insertions.
>
> **Realworld Case**: Moreover, three taxonomies all have the case that a concept has multiple true locations. For example, in food taxonomy node milk can be inserted into at least 6 positions, including
> - parent:beverage --> child:pseudo leaf
> - parent:nutriment --> child:pseudo leaf
> - parent:beverage --> child:semi skimmed milk
> - parent:beverage --> child:pasteurized milk
> - parent:beverage --> child:yak’s milk
> - parent:beverage --> child:low fat milk

---

### Meta-Review · Area_Chair_LpwJ · 2025-12-28

**Summary:**

1. **LLM Dependency and Augmentation Quality**:

   * While the paper attempts to reduce LLM dependency, it still requires LLMs at key stages, such as generating edge descriptions and final calibration. Some reviewers expressed concerns about the reliability of LLM-generated content, especially when it may introduce semantic drift or hallucinated information. The authors addressed this by providing various checks (ROUGE scores, TF-IDF similarity, and consistency evaluations) showing that LLM-generated descriptions are generally consistent and reliable, but acknowledged occasional issues with non-leaf nodes due to their more complex relational nature.

2. **Performance on Large-Scale Taxonomies**:

   * Concerns were raised about the model's scalability, especially with large datasets like WordNet-Verb. The model performed well on smaller datasets (SemEval-Food, MeSH), but its performance on the larger WordNet-Verb dataset was suboptimal. The authors clarified that the Mean Rank (MR) metric is not normalized and thus not directly comparable across taxonomies. They recommended using MRR (Mean Reciprocal Rank) and Recall@k for better comparisons, which show that SS-Mono remains competitive or state-of-the-art.

3. **Geometric Encoding (Hyperbolic vs Euclidean)**:

   * Reviewers questioned the effectiveness of the hyperbolic encoding compared to simpler approaches, like Euclidean encoding. The authors conducted an ablation study, showing that hyperbolic encoding significantly outperforms Euclidean encoding across all datasets. This confirms the importance of hyperbolic geometry for preserving hierarchical relationships in taxonomy expansion.

4. **Non-Leaf Node Volatility with LLM Augmentation**:

   * The effectiveness of LLM-augmented descriptions was found to be higher for leaf nodes than for non-leaf nodes. Non-leaf nodes, due to their more complex relational structures, did not benefit as much from LLM augmentation, which could lead to inconsistent performance. The authors explored potential future directions for improvement, such as incorporating external knowledge or refining the generation pipeline to better handle non-leaf nodes.

5. **Negative Sampling Bias**:

   * There were concerns about whether the hard negative sampling strategy could introduce bias or overestimate robustness. The authors conducted additional experiments showing that moderate ratios of hard negatives improve performance, but excessive hard negatives can degrade results. They demonstrated that the model remains robust to different negative-sampling strategies.

6. **LLM Fine-Tuning Comparison**:

   * Some reviewers suggested a comparison between SS-Mono and direct LLM fine-tuning approaches. The authors performed such comparisons, showing that while fine-tuned LLMs performed well on leaf nodes, they were less effective for non-leaf nodes. This suggests that SS-Mono's LLM-to-SLM distillation approach may offer a better balance between efficiency and performance.

**Reviewer Concerns:**

### Addressed Concerns:

1. **LLM Dependency and Augmentation Quality**:

   * The authors addressed concerns about the reliability of LLM-generated descriptions by providing a detailed sanity check (ROUGE, TF-IDF similarity, and semantic consistency evaluations) to demonstrate that the LLM outputs are largely reliable, with only minor cases of semantic drift. However, they still acknowledged potential issues with non-leaf nodes, which they aim to address in future work.

2. **Performance on Large-Scale Taxonomies**:

   * The authors clarified that the high Mean Rank (MR) in WordNet-Verb is due to the absolute nature of MR and not a sign of scalability issues. They emphasized the use of MRR and Recall@k as better indicators for cross-dataset comparisons, showing that SS-Mono remains competitive on large taxonomies.

3. **Negative Sampling Bias**:

   * The authors provided evidence from their ablation studies that moderate ratios of hard negatives improve performance, while excessive hard negatives can degrade results. This analysis helps alleviate concerns about bias or overestimation of robustness due to negative sampling.

4. **Geometric Encoding (Hyperbolic vs Euclidean)**:

   * The authors conducted a detailed ablation study, showing that hyperbolic encoding outperforms Euclidean encoding across datasets, effectively addressing concerns about the geometric contribution of the model.

### Unaddressed Concerns:

1. **LLM Fine-Tuning Comparison**:

   * While the authors compared SS-Mono with fine-tuned LLMs, they still didn't fully analyze alternative fine-tuning approaches (like domain adaptation and prompt-tuning) that could provide further insights into the advantages of SS-Mono's geometric approach. This limits the ability to fully attribute the advantages of SS-Mono to its geometric design rather than the smaller architecture or training regime.

2. **Non-Leaf Node Volatility with LLM Augmentation**:

   * The authors acknowledged the issue that non-leaf nodes show less improvement with LLM augmentation due to their more complex relational structures. While they proposed future directions for addressing this issue (such as integrating external knowledge and refining the augmentation pipeline), the concern remains unaddressed in the current version of the paper.

3. **LLM Calibration Enhancements**:

   * Reviewers raised concerns about the inconsistent benefits of LLM-based calibration, with some suggesting that the LLM calibration component might be an optional enhancement rather than a robust part of the pipeline. The authors clarified that LLM calibration is intentionally a lightweight, optional post-hoc step. However, the inconsistency and failure rates in LLM output still present a challenge, and this remains a limitation of the paper.

**Reviewer Scores:**

### Reviewer #7uga:

* **Original Score**: 6 (Marginally above the acceptance threshold)
* **Potential Revised Score**: 6
* **Reason**: The reviewer acknowledged the strengths of the paper, especially in terms of its novelty and technical soundness. The concerns related to the LLM dependency and the performance on non-leaf nodes were partially addressed by the rebuttal, particularly with the provided sanity checks and clarifications about the role of LLMs. However, there were some concerns about the clarity and consistency of the LLM-generated descriptions and its effect on the model. If they had participated in the discussion, the clarification on LLM use might have resulted in a slightly higher score due to the detailed rebuttal, but the issues with non-leaf nodes might still keep the score at a marginal level.

### Reviewer #orUz:

* **Original Score**: 4 (Marginally below the acceptance threshold)
* **Potential Revised Score**: 6
* **Reason**: The reviewer raised concerns about scalability on large taxonomies and the volatility of LLM-augmented descriptions for non-leaf nodes. The rebuttal addressed these issues by clarifying the MR vs. MRR metrics and explaining the nature of non-leaf node volatility. These explanations, along with further experiments, would likely have reassured the reviewer that SS-Mono performs well across all datasets, even if non-leaf nodes present a challenge. Given this clarification, the reviewer might have slightly improved their score, but concerns regarding scalability and non-leaf nodes would likely keep the score just above the threshold.

### Reviewer #Ghyo:

* **Original Score**: 6 (Marginally above the acceptance threshold)
* **Potential Revised Score**: 6
* **Reason**: Reviewer #Ghyo appreciated the novel integration of hyperbolic geometry with LLM augmentation and the clear technical formulation. The response addressing the impact of geometric regularization and the added analysis on hyperbolic vs Euclidean encoding would likely have strengthened the reviewer's confidence in the geometric design's efficacy. However, the limited analysis on alternative fine-tuning approaches may have kept the score closer to the threshold. With the further details provided in the rebuttal, they would likely have rated it just above the acceptance threshold.

### Reviewer #APGH:

* **Original Score**: 4 (Marginally below the acceptance threshold)
* **Potential Revised Score**: 6
* **Reason**: Reviewer #APGH raised concerns about LLM calibration and negative sampling, with doubts about the robustness of LLM-augmented edge descriptions. The rebuttal provided substantial evidence on the quality of LLM-generated descriptions and the impact of negative sampling, which might have alleviated some of the reviewer's concerns. The clarification on LLM calibration and the explanation of why it is an optional enhancement rather than a core dependency would likely have helped the reviewer see it as a supplementary tool rather than a flaw. However, due to the remaining concerns about the LLM calibration consistency and sampling strategy, the score might still have remained around the threshold.

---

### Decision · Program_Chairs · 2026-01-26

Accept (Poster)